# A tissue-specific, Gata6-driven transcriptional program instructs remodeling of the mature arterial tree

Marta Losa[1], Victor Latorre[1], Munazah Andrabi[1], Franck Ladam[2], Charles Sagerström[2], Ana Novoa[3], Peyman Zarrineh[1], Laure Bridoux[1], Neil A Hanley[1,4], Moises Mallo[3]*, Nicoletta Bobola[1]*

[1]Faculty of Biology, Medicine and Health, University of Manchester, Manchester, United Kingdom; [2]Department of Biochemistry and Molecular Pharmacology, University of Massachusetts Medical School, Worcester, United States; [3]Instituto Gulbenkian de Ciência, Oeiras, Portugal; [4]Endocrinology Department, Central Manchester University Hospitals NHS Foundation Trust, Manchester, United Kingdom

**Abstract** Connection of the heart to the systemic circulation is a critical developmental event that requires selective preservation of embryonic vessels (aortic arches). However, why some aortic arches regress while others are incorporated into the mature aortic tree remains unclear. By microdissection and deep sequencing in mouse, we find that neural crest (NC) only differentiates into vascular smooth muscle cells (SMCs) around those aortic arches destined for survival and reorganization, and identify the transcription factor Gata6 as a crucial regulator of this process. Gata6 is expressed in SMCs and its target genes activation control SMC differentiation. Furthermore, Gata6 is sufficient to promote SMCs differentiation in vivo, and drive preservation of aortic arches that ought to regress. These findings identify Gata6-directed differentiation of NC to SMCs as an essential mechanism that specifies the aortic tree, and provide a new framework for how mutations in GATA6 lead to congenital heart disorders in humans.
DOI: https://doi.org/10.7554/eLife.31362.001

*For correspondence: mallo@igc.gulbenkian.pt (MM); nicoletta.bobola@manchester.ac.uk (NB)

Competing interests: The authors declare that no competing interests exist.

## Introduction

In mammals, effective delivery of oxygenated blood from the heart to the systemic circulation relies on physical separation from deoxygenated blood in the pulmonary circulation. This separation is contingent upon two critical developmental events: cardiac septation, including division of the outflow tract (OFT) into the ascending aorta and the pulmonary trunk (*Hutson and Kirby, 2007*; *Kirby and Hutson, 2010*); and complex remodeling of the embryonic circulation. In mid embryogenesis, the aortic arches (AAs), five bilaterally symmetric embryonic vessels, traverse the branchial arches and distribute blood from the heart to the dorsal aortae. During remodeling, which in mouse occurs between embryonic day (E) 9.5 and E13.5 (*Hiruma et al., 2002*), AA1 and AA2 regress into capillary beds, while asymmetric partial regression of the right-sided vessels leads to AA3-6 being incorporated into the major thoracic arteries (the carotids, aortic arch and subclavian arteries). Failure to separate pulmonary and systemic circulations in this manner results in congenital heart disease characterized by cyanosis, due to the mixing of deoxygenated with oxygenated blood (*Hutson and Kirby, 2007*; *Keyte and Hutson, 2012*).

This need for separation and the consequences of cyanosis have made understanding the mechanism(s) of OFT septation and AA remodeling a high priority. In addition to haemodynamics (*Yashiro et al., 2007*), cardiac neural crest (NC) cells, a subpopulation of the cranial NC, are known

to be essential. These cells delaminate from the posterior hindbrain and colonize the branchial arches containing AA3-6 AAs or continue to the OFT (*Boot et al., 2003*; *Waldo and Kirby, 1993*). At destination, cardiac NC cells differentiate into vascular smooth muscle cells (SMC), a type of SMC found in the walls of large and medium sized vessels, mainly arteries (*Jiang et al., 2000*). Although AAs initially form, ablation of cardiac NC cells in chick or mouse embryos causes their regression and blocks separation of the pulmonary trunk and aorta, leading to severe cyanosis [persistent truncus arteriosus (PTA)] (*Kirby and Stewart, 1983*; *Porras and Brown, 2008*; *Waldo et al., 1996*). Genetic inactivation has also identified individual factors or pathways required for correct remodelling and separation of the systemic and pulmonary circulation, such as Notch, required to generate SMCs, Tgfß or endothelin signaling (*High et al., 2007*; *Kurihara et al., 1995*; *Manderfield et al., 2012*; *Molin et al., 2002*; *Wang et al., 2006*; *Yanagisawa et al., 1998*); Sema3c and its receptors, Nrp1 and Plxnd1 (*Feiner et al., 2001*; *Gitler et al., 2004*; *Kawasaki et al., 1999*); or the transcription factor (TF) Gata6 (*Kodo et al., 2009*; *Lepore et al., 2006*). However, at present, the connection between these genetic factors and how cardiac NC cells regulate development and separation of the circulations, and their relative importance versus blood flow is largely unknown.

By combining microdissection and global transcriptomes analysis, we identify the transcriptional signature underlying NC differentiation to vascular SMCs, uniquely in the aortic arches destined for reorganization into the mature aortic tree. Computing TFs binding levels predicts GATA as main transcriptional regulators in the areas colonized by cardiac NC. We show that Gata6 is sufficient to promote NC differentiation to SMCs in vivo, and also to preserve AAs that normally regress. Our work identifies Gata6 as a crucial regulator of vessel fate and vascular remodelling, both of which are critical for normal development of the mature aortic tree.

## Results

### Identification of an active SMC differentiation program in the posterior branchial arches and OFT

To identify the molecular pathways underlying development of the adult great vessels, we examined gene expression in posterior branchial arches (PBA) and OFT at two defined developmental stages, E10.5 (Theiler stage, TS17) and E11.5 (TS19), by RNA-seq analysis (*Figure 1A* and *Figure 1—figure supplement 1A*). The last of the AAs to be formed (sixth) becomes evident at E10.5, and the organization of the branchial AAs is still highly symmetrical at both stages (*Hiruma et al., 2002*). To identify transcripts specifically enriched in this embryonic area, we compared these datasets with the transcriptome of the second branchial arch (BA2, *Figure 1A*), which is similarly colonized by NC, but does not contribute to any of the mature great vessels. Expression profiles display the highest variation across tissues already at the earliest time point examined, shortly after cardiac NC populates the PBA/OFT (*Figure 1B*; *Figure 1—figure supplement 1B*). After data normalization, we extracted differentially expressed (DE) genes between BA tissues (*Supplementary file 1*). DE genes largely consist of transcripts enriched in the PBA/OFT, and a large fraction of those is shared between early and late stages (*Figure 1C*). Consistent with the contribution of PBA/OFT to the heart circulation, PBA/OFT-enriched transcripts were associated with 'heart development', 'OFT morphogenesis' and 'vasculogenesis' (*Figure 1D*). On closer inspection, we found that a considerable proportion of genes contained in the gene ontology (GO) 'heart development' is associated with AAs and/or OFT phenotypes (36%; 12 out of 33). PBA/OFT-enriched genes also included transcripts highly expressed in the myocardium, owing to the OFT component (*Figure 1—figure supplement 1C*). Among the transcripts displaying the most significant changes, we found *Acta2*, *Tagln*, *Cnn1* and *Myocd*, which are highly expressed in SMCs (*Figure 1E*; *Figure 1—figure supplement 1C*). *Myocd* encodes for a TF that is sufficient to activate the program of SMC differentiation (*Li et al., 2003*; *Wang et al., 2003*) and functions in complex with the MADS box TF serum response factor (SRF). Intriguingly, a highly significant fraction of transcripts enriched in PBA/OFT at both developmental stages corresponds to SRF-responsive genes (*Figure 1F*). In sum, our comparative analysis indicates that a transcriptional program controlling the generation of SMCs is specifically activated in the PBA/OFT.

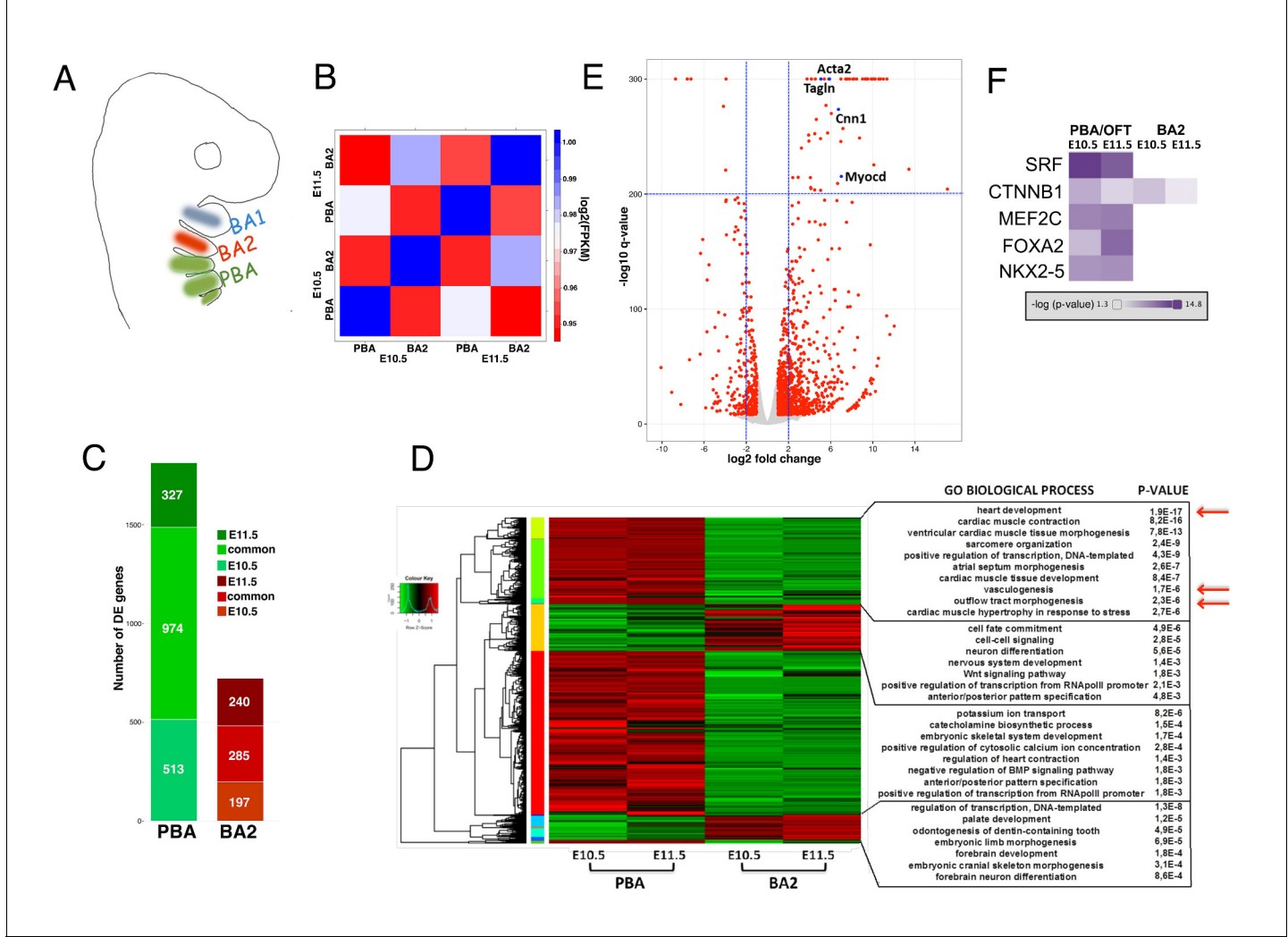

**Figure 1.** Global transcriptomes of developing BAs and OFT. (**A**) In mammals, BA2 (red) hosts aortic arch (AA)2, which regresses, while posterior BAs (PBA, green) host AA3-6, which contribute to the main thoracic arteries. (**B**) Correlation plot of global expression profiling separates BA2 and PBA at E10.5 and E11.5. Heatmap shows the Spearman correlation coefficients for each pair of samples based on the normalized expression values (FPKM). (**C**) Count of DE genes between PBA and BA2 (fold change ≥±2; q-value <0.05) at E10.5–11.5. The bar plot shows PBA/OFT-enriched genes as the largest fraction of DE genes (color-coded as in A). (**D**) Hierarchical clustering of DE genes in any of the three pair-wise comparisons. DAVID analysis (*Huang et al., 2009*) of the clusters generated detects significant association with PBA/OFT-specific Biological Process GO terms (arrows). (**E**) Volcano plot. Top significant PBA/OFT-enriched genes are highly expressed in SMCs. (**F**) Ingenuity Pathway Analysis (IPA) identifies the TF SRF as the most likely upstream regulator of PBA/OFT-enriched genes (p-value<1e-20). SRF targets include *Myocd* and known Myocd targets (e.g. *Acta2, Tagln*).
DOI: https://doi.org/10.7554/eLife.31362.002

The following figure supplement is available for figure 1:

**Figure supplement 1.** Differential expression across developing BAs.
DOI: https://doi.org/10.7554/eLife.31362.003

## Differentiation of SMCs in vivo is restricted to PBA/OFT

Transcriptomic analyses revealed a high enrichment in transcripts related to the process of SMCs differentiation in the PBA/OFT relative to BA2. Therefore, we analyzed the presence of SMCs in the BA region of mouse embryos. BAs are traversed by five embryonic AAs, which connect the heart with the paired dorsal aortae and allow blood circulation throughout the embryo. AAs develop in a rostro-caudal sequence, beginning around E8.5 with formation of the AA1, which traverses BA1 (*Hiruma et al., 2002*; *Waldo and Kirby, 1998*). Within the next two days, AAs 2, 3, 4 and 6 (none of the arteries is named as 5) appear sequentially. Using confocal immunofluorescence analysis, we

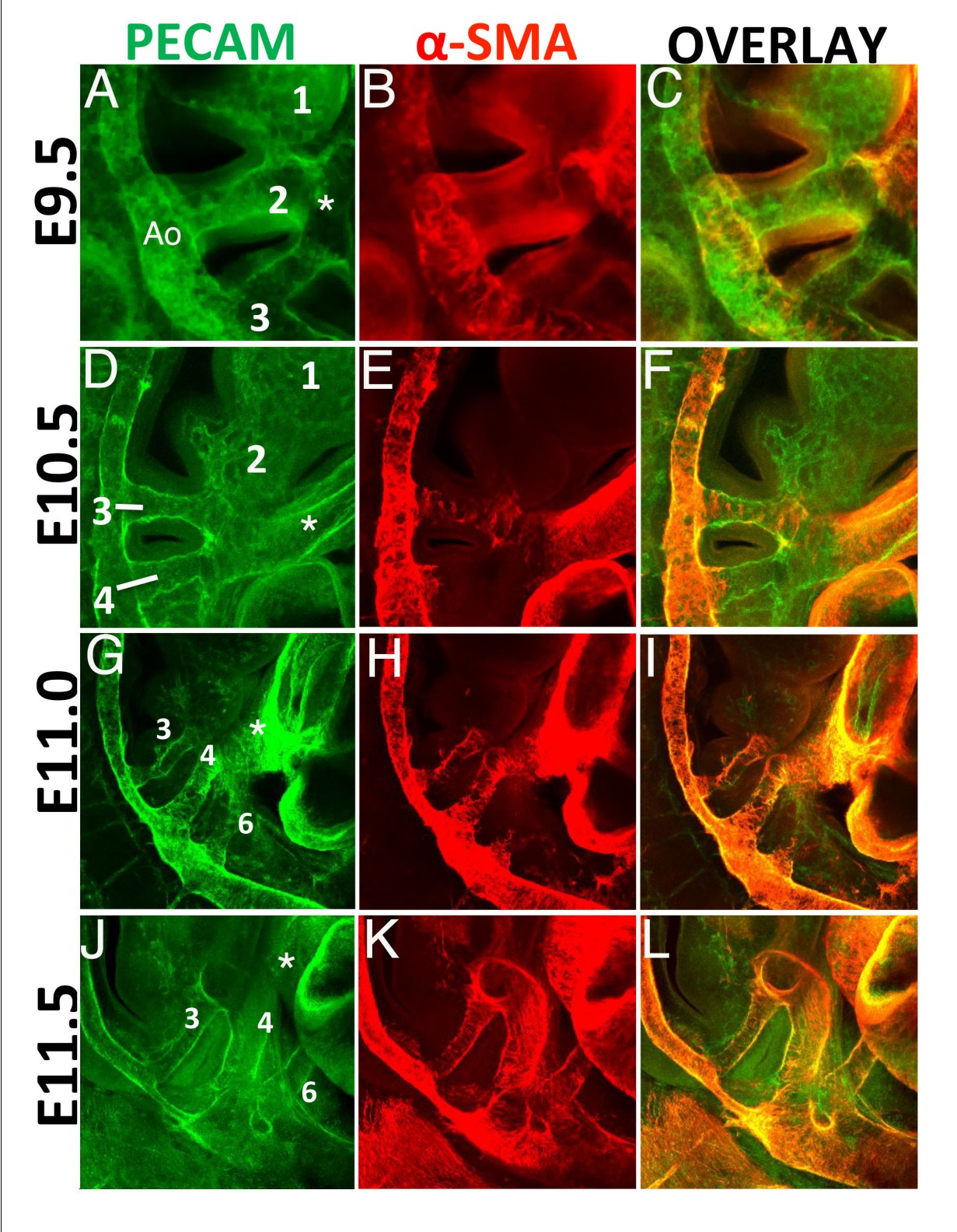

**Figure 2.** SMCs are generated exclusively in BA3-6, in a rostro-caudal fashion. Confocal analysis of whole mount wild type embryos at E9.5 (**A–C**), E10.5 (**D–F**), E11.0 (**G–I**) and E11.5 (**J–L**) to visualize the endothelial network (PECAM, green) (**A,D,G, J**) and vascular SMCs (SMA, red) (**B, E, H, K**). C, F, I, L are overlays of PECAM and SMA staining. (**A-C**) At E9.5 (25 somites, (**s**), AAs 2 and 3 are connected to the dorsal aorta. AA1 has already started to regress into capillary beds. SMA-positive cells are visible in the dorsal aorta and OFT, but are not detected in the walls of AA2-3. (**D–F**) At E10.5 (37s), AA3 and

*Figure 2 continued on next page*

Figure 2 continued

AA4 are fully formed, but only AA3 is associated to vascular SMCs. Capillary beds replace AA2 in BA2. G–I. A clear AA6 is visible at E11.0 (43s). At this stage AA3-4 are fully covered with SMA-positive cells, while the newly formed AA6 is not associated with vascular SMCs. (J–L) At E11.5 (48s) formation of AA3, 4 and 6 is complete. All visible AAs (AA3-4-6) are covered by SMA-positive cells. Right view. Ao, aorta; asterisks label the OFT.

DOI: https://doi.org/10.7554/eLife.31362.004

observed AA2 and AA3 at E9.5 (*Figure 2A*). We never observed AA1 at this stage, presumably because this vessel, which is the first to form, had already regressed (*Hiruma et al., 2002*). One day later (E10.5), we detected well-formed AA3 and AA4, both connected to the aortic sac/OFT and the dorsal aorta (*Figure 2D*). Consistent with the regression process of AA1-2, PECAM staining revealed capillary beds in the BA1-2 instead of the arteries observed in younger embryos. At later stages (E11.0-E11.5), three vessels were present in the BA region, AAs 3, 4 and 6, and regression of the two rostral-most AAs was complete (*Figure 2G,J*).

We next examined the distribution of vascular SMCs in these vessels. At E9.5, none of the AAs was associated with vascular SMCs, but we detected vascular SMCs in the dorsal aorta and the OFT (*Figure 2B,C*). At E10.5, we observed the presence of vascular SMCs in AA3 (*Figure 2E,F*), but not in the developing AA4. At E11.0, AA3 and AA4 were fully covered by vascular SMCs (*Figure 2H,I*) and half a day later, all posterior AAs were associated with vascular SMCs (*Figure 2K,L*). These results indicate that the three most posterior AAs (3-6), which will contribute to the mature aortic tree, recruit vascular SMCs to their walls shortly after their initial formation, in a rostro-caudal progression. In contrast, the two most anterior AAs (1-2) never become associated with vascular SMCs. Thus, the distribution of SMCs correlates with vessels selected for preservation. In sum, the process of SMC differentiation specifically takes place in PBA/OFT, consistent with our transcriptomic analyses.

## Identification of the main regulators of the PBA/OFT transcriptional network

We devised an unbiased approach to identify main transcription factors (TF) driving gene expression in PBA/OFT. Binding of multiple TFs, or collaborative DNA binding, can counteract nucleosome repositioning, facilitates access of each TF to DNA and leads to increased binding levels (*Biggin, 2011*; *Spitz and Furlong, 2012*). We reasoned that changes in the binding levels of 'ubiquitous' TFs, combined with sequence analysis of the underlying DNA, could be used to detect tissue-specific, combinatorial TFs occupancy in the PBA/OFT. We analyzed binding of Meis TFs, which regulate development of many diverse organs, including eye, heart and limb (*Capdevila et al., 1999*; *Mercader et al., 2000*; *Stankunas et al., 2008*; *Zhang et al., 2002*). Meis1 (*Figure 3A*) and Meis2 (not shown) are expressed in both BA2 and PBA and bind broadly and to largely overlapping regions in the BAs (*Amin et al., 2015*). Further supporting a widespread role for Meis TFs, Meis occupy up to 60% accessible chromatin in the BA2 (*Minoux et al., 2017*) (*Figure 3—figure supplement 1A*). Moreover, Meis recognition motifs are overrepresented in cardiac enhancers and TBX5 and NKX2–5oc cupied regions in cardiomyocytes (*Luna-Zurita et al., 2016*; *Wamstad et al., 2012*). As expected (*Amin et al., 2015*), we found that Meis binding sites in the PBA/OFT are largely shared with BA2 (*Figure 3B,C*). Using DiffRep (*Shen et al., 2013*), we identified 1379 regions at which Meis binding level was highly increased (LogFC $\geq$3) in the PBA/OFT vs BA2 (*Figure 3B,C*; *Supplementary file 2*). Functional annotation of Meis highly occupied regions in the PBA/OFT revealed association with PBA-specific biological processes, such as 'regulation of SMC differentiation', and 'regulation of vasculature development' (*Figure 3D*), supporting the validity of this approach. We hypothesized that interrogation of DNA sequence at sites with increased Meis binding signal would reveal the identity of major TFs driving PBA/OFT gene expression. The most highly overrepresented motifs identified by de novo motifs analysis, matched recognition sequences for GATA, Forkhead, HAND and TEAD, together with motifs recognized by Meis and Meis partners Hox/Pbx (*Figure 3E*). The subset of Meis-bound regions containing GATA motifs were associated with 'blood vessel morphogenesis', while regions containing Ebox and Forkhead motifs were mainly associated with striated and cardiac muscle development (*Figure 3—figure supplement 1B,C*). Strikingly, *Gata6* conditional inactivation in the cranial NC and in SMC results in abnormal

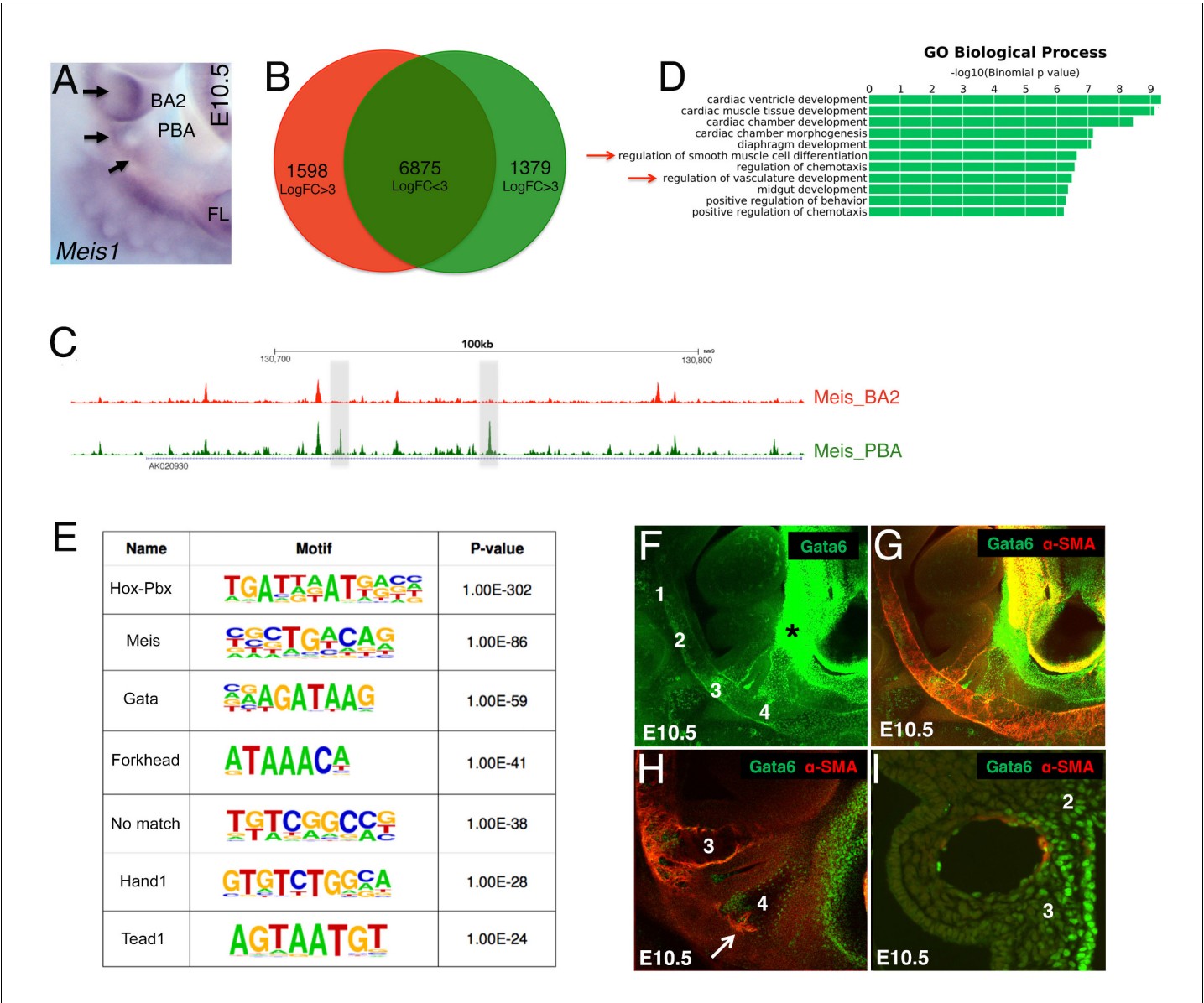

**Figure 3.** Unbiased identification of main TFs instructing PBA-specific transcription. (A) Whole mount in situ hybridizations on E10.5 mouse embryo. *Meis1* is expressed in the BA2 (arrow) and in the PBA (BA3 and BA4, arrows). (B) Diffrep analysis of Meis binding in PBA and BA2. Venn diagram shows 'shared' Meis binding sites (FE ≥ 10) with logFC < 3 signal (n = 6875) and with higher (logFC ≥ 3) signal in PBA (green) and BA2 (red). (C) UCSC browser tracks upstream of *Nrp1* illustrate the largely overlapping binding of Meis in PBA and BA2, with instances of increased Meis binding signal (gray regions) in PBA relative to BA2. (D) Top over-represented biological processes associated to Meis peaks (FE ≥ 10) with higher signal in PBA (logFC ≥ 3) include 'smooth muscle cell differentiation' and 'vasculature development'. (E) De novo motif discovery on Meis peaks (FE ≥ 10) with higher signal in PBA identifies high enrichment of the recognition sequence for GATA TFs, together with motifs recognized by Meis and Meis partners Hox/Pbx. (F–I) Confocal analysis of E10.5 embryos (36–37 s) whole mount (F–H) and sagittal sections (I) stained with Gata6 (F) and Gata6/SMA antibodies (G–I). Gata6 is detected in the BA3, BA4 (F–H) and the heart region (F–G). Individual z-stack (H) shows Gata6 co-localization with SMA-positive cells in the BA3 and the appearance of SMCs in Gata6-positive area in the BA4 (arrow). Gata6/SMA-positive cells surround AA3 (I). Numbers indicate corresponding BAs; asterisks label the OFT.

DOI: https://doi.org/10.7554/eLife.31362.005

The following figure supplement is available for figure 3:

**Figure supplement 1.** Analysis of Meis-bound regions containing different TF recognition motifs.
DOI: https://doi.org/10.7554/eLife.31362.006

development of the great vessels and OFT (*Kodo et al., 2009*; *Lepore et al., 2006*). Consistent with changes in combinatorial occupancy reflecting changes in protein concentration, *Gata6*, together with *Gata4* and *Gata5*, was among the most significantly enriched transcripts in PBA/OFT (*Figure 1—figure supplement 1C*). Next, we analyzed Gata6 expression in the BA region. In agreement with our RNA-seq analyses, we found a striking dichotomic distribution of Gata6, with high enrichment in the PBA and OFT, relative to BA1-2 (*Figure 3F,G*). Consistent with a role in initiating SMC differentiation in the BAs, Gata6 was detected in vascular SMCs associated with AA3 (*Figure 3F–I*), and in BA4 (*Figure 3H*), just before SMC differentiation around AA4 *Figure 3F–H*).

## Gata6 binding is associated with transcription of genes involved in vascular development

To explore the role of Gata6, we examined its genomic localization in the PBA/OFT tissue at E11.5. Using chromatin immunoprecipitation followed by next-generation sequencing (ChIP-seq), we identified 5122 bound regions with fold enrichment (FE) $\geq$ 10 (*Supplementary file 3*). Gata6 peaks are mostly distributed in intergenic and intronic regions, and a significant percentage overlaps promoters (*Figure 4A* and *Figure 4—figure supplement 1A*). We observed high Gata6 peaks at *Gata6* and *Gata4* loci, which suggest that GATA TFs regulate their own genes in the PBA/OFT (*Figure 4B*). Accordingly, in E12.5 heart ventricles, Gata4 binds at the same locations in the *Gata6* gene (*He et al., 2014*). To characterize Gata6 binding, we performed de novo motif discovery on Gata6 peak summits (200nt). The most significantly enriched motif matched motifs characterized previously for members of the GATA family (*DeVilbiss et al., 2016*) (*Figure 4C*), with >90% of bound regions containing at least one GATAR consensus binding sequence. The second top motif was the previously identified recognition sequence for the TEAD family of TFs (*Figure 4—figure supplement 1B* and *Figure 3E*). Functional annotation of Gata6 peaks using GREAT (*McLean et al., 2010*) identified almost exclusively cardiovascular-related GO terms, such as 'heart and OFT morphogenesis' and 'artery morphogenesis' (*Figure 4—figure supplement 1C*). Strikingly, and supporting a regulatory activity for Gata6 binding, Gata6 peaks clustered around genes whose mutations cause defects that mirror *Gata6* loss of function phenotypes in the PBA/OFT, such as 'abnormal outflow tract development' and 'heart and great artery attachment' (*Figure 4D*). Next, we asked whether Gata6 binding functions to control gene expression. We first generated independent genome-wide maps of H3K27ac in E11.5 PBA/OFT and BA2 to delineate active enhancers and promoters (*Creyghton et al., 2010*). Tissue-specific H3K27ac signatures at promoters and distal sites correlated with gene expression differences between PBA/OFT and BA2 (*Figure 4E*), confirming that H3K27Ac decorates regions actively engaged in promoting gene expression. Regions marked by H3K27ac in the PBA/OFT overlapped with a large fraction of Gata6 sites (42%) (*Figure 4F* and *Supplementary file 3*). Gata6 sites had the highest degree of overlap with PBA-specific enhancers (820 PBA-specific and 98 BA2-specific regions) (*Figure 4G*), which were also significantly associated with DE genes enriched in the PBA/OFT (24.5% of Gata6-bound enhancers were associated with PBA/OFT-enriched genes compared to 11.1% for the entire set of acetylated regions in the PBA; p-value=2.2e-16). These observations suggest that Gata6 binds to distal enhancers and activates gene expression. Notably, active enhancers bound by Gata6 were significantly associated with PBA-enriched genes, involved in 'vasculature' and 'blood vessel development' (*Figure 4—figure supplement 1D*); these include components of the Notch signaling pathway (*Notch1, Hes1* and *Jag1*), which is essential for cardiac NC differentiation to SMCs (*High et al., 2007*; *Manderfield et al., 2012*), and the master regulator of SMC differentiation *Myocd* (*Wang et al., 2003*). Promoters associated with active enhancers bound by Gata6 were also significantly enriched in GATA and TEAD recognition motifs, further suggesting that the GATA/TEAD module is a regulatory feature of the PBA/OFT transcriptional network (*Figure 4—figure supplement 1E*). We detected prominent Gata6 peaks at *Jag1* and *Myocd* loci (*Figure 4H,I*), contained in regions highly acetylated in PBAs but not acetylated in BA2, as expected for active, tissue-specific enhancers (henceforth referred to as *Jag1* Cis Regulatory Element (CRE) one and *Myocd* CRE 1–2; VISTA was reported to drive heart-specific activity (*Visel et al., 2007*). When tested in zebrafish, *Jag1* CRE1 and *Myocd* CRE1 and CRE2 displayed high activity in the heart region (*Figure 4H,I* and *Figure 4—figure supplement 2A,C*). Each CRE contains at least two GATA recognition motifs (*Figure 4—figure supplement 2B*). By introducing mutations in the GATA consensus motifs (*Figure 4—figure supplement 2B,C*), we detected a highly significant reduction in the activity of the mutant *Jag1* CRE1 and *Myocd* CRE2 GFP reporters

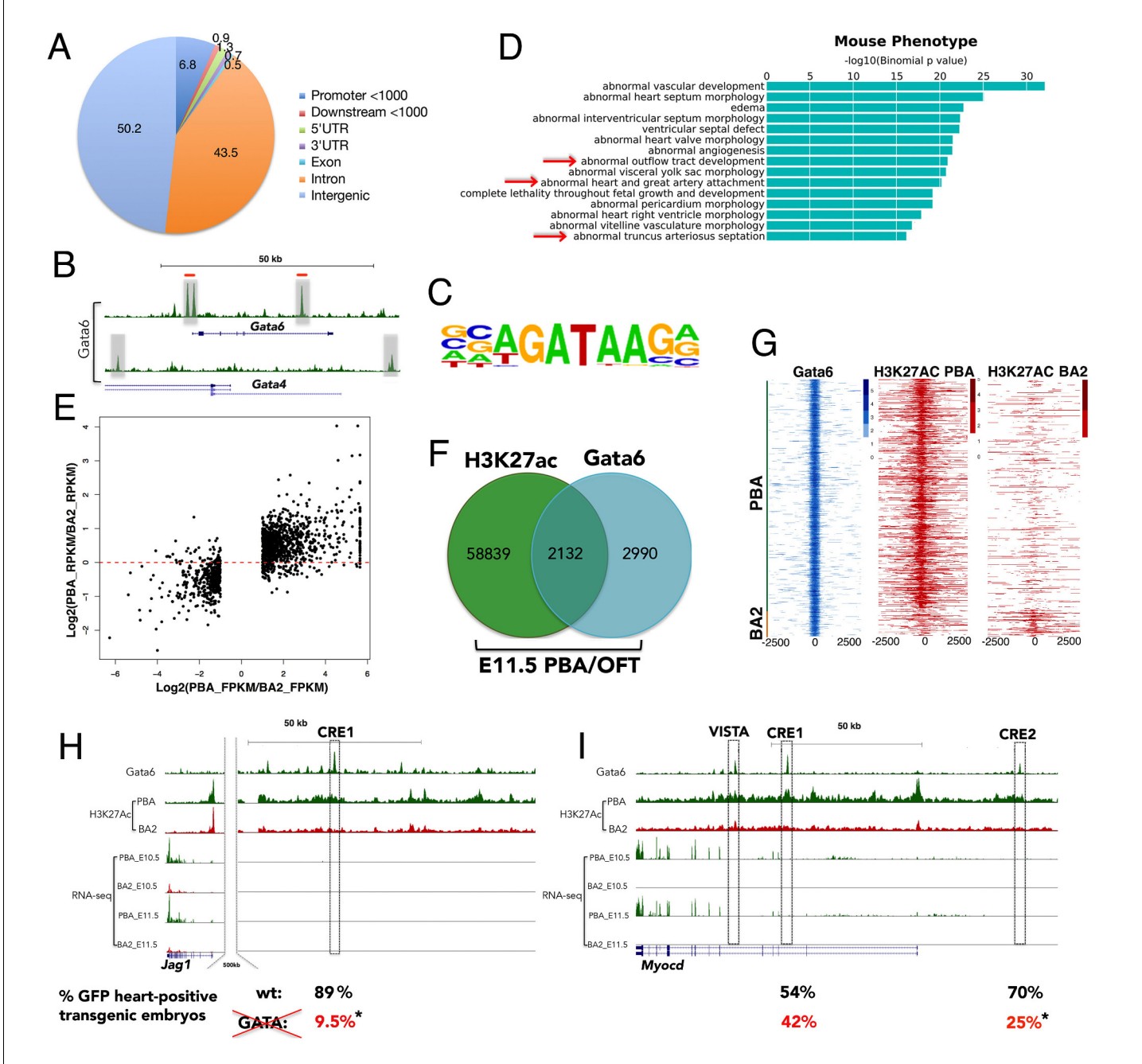

**Figure 4.** Gata6 occupies active enhancers in PBA/OFT. (**A**) CEAS analysis of the distribution of Gata6 peaks relative to Reference Sequence (RefSeq) gene structures. The pie chart and corresponding percentage values indicate the proportion of reads. (**B**) UCSC browser tracks shows Gata6 binding (gray regions) at *Gata4* and *Gata6* loci. Red lines indicate Gata4 binding in E12.5 ventricles. (**C**) Sequence logo of the most significant motifs identified using de novo motif discovery. (**D**) Top over-represented mouse phenotypes associated to Gata6 peaks. The x axes values correspond to the binomial raw (uncorrected) p-values. (**E**) Scatterplot of the Log2(ratio of FPKM) values for DE genes between PBA and BA2 versus the Log2(ratio of RPKM) values of their H3K27ac signals at promoters and distal regions (Correlation: 0.694; p<2.2e-16). (**F**) Venn diagram (not proportional) of Gata6 peaks (200nt summits) and H3K27Ac-positive regions in the PBA/OFT. Almost half of Gata6 peaks overlap regions acetylated in the PBA/OFT. (**G**) Heatmap of Gata6 peaks and corresponding H3K27ac peaks (within 5000 nt of the summit) shows most Gata6 peaks overlap PBA/OFT-specific enhancers (n = 820), and only a minority of BA2-specific enhancers (n = 98). Regions detected as H3K27Ac-positive in both PBA/OFT and BA2 were excluded from this analysis. (**H, I**) UCSC browser tracks. RNA-seq and ChIP-seq profiles for Gata6 binding and H3K27Ac in the PBA/OFT and H3K27Ac in the BA2 at *Jag1* (**H**) and *Myocd* (**I**) loci. Gata6 binds regions highly acetylated in PBA/OFT, but not BA2 (boxed; VISTA highlights a heart-positive enhancer). Numbers

*Figure 4 continued on next page*

*Figure 4 continued*

correspond to the % of embryos injected with wild-type (upper row, in black) and mutant (lower row, in red) enhancers, displaying reporter activity in the heart in addition to the midbrain (positive control); asterisks denote p-value<0.05 (Fisher's Exact Test).

DOI: https://doi.org/10.7554/eLife.31362.007

The following figure supplements are available for figure 4:

**Figure supplement 1.** Extended analysis of Gata6 peaks.
DOI: https://doi.org/10.7554/eLife.31362.008

**Figure supplement 2.** Disrupting GATA motifs affects the activity of GATA6-bound enhancers.
DOI: https://doi.org/10.7554/eLife.31362.009

in developing zebrafish embryos, demonstrating that direct binding of GATA TFs to these elements instructs their tissue-specific activity (*Figure 4H,I* and *Figure 4—figure supplement 2C–E*). In sum, Gata6 binding is associated with the active, tissue-specific regulation of known target genes involved in vasculature development. Gata6 binds to *Jag1* and *Myocd* enhancers, whose activity requires intact GATA motifs, suggesting that Gata6 directly controls *Jag1* and *Myocd* transcription. Regulation of these transcripts in combination would be expected to promote development of the AAs into mature blood vessels, equipped with a tunica media of SMCs.

## Forced expression of GATA6 in the neural crest generates SMCs

Gata6 is specifically expressed in the PBA/OFT, as vascular SMCs are recruited to the AAs, and Gata6 binding is associated with tissue-specific transcription of genes controlling SM and vascular development. Next, we asked if GATA6 is sufficient to promote vascular SMC differentiation in the BAs. To force *Gata6* expression in NC cells migrating into anterior BAs, we created transgenic embryos expressing *Gata6* under the control of the *Hoxa2* enhancer, which drives expression in the cranial NC directed to BA2 (*Nonchev et al., 1996*). We imaged a subset of *a2::Gata6* embryos (7 out of 11) to monitor SMC differentiation, and observed ectopic SMCs in the BA2 of all embryos expressing *Gata6* in this BA at E10.5 (5/5) (5D and *Figure 5—figure supplement 1D,E*). Small clusters of ectopic SMCs were also visible along the NC migration route, before ingression into BA2 (*Figure 5—figure supplement 1E*, asterisk). While dissecting, we observed blood in the BA2 in the majority of transgenic embryos, a sign of a patent artery. Indeed, we observed persistent AA2 in the majority of *a2::Gata6* embryos with ectopic Gata6 and SMCs in the BA2 at E10.5 (3/5; *Figure 5E*; *Figure 5—videos 1–4*). Conversely, we did not detect any SMCs in BA2 of control littermates (n = 11; *Figure 5A* and *Figure 5—figure supplement 1B*) or unrelated wild-type embryos analyzed (n > 30), and in all these cases the AA2 had completely regressed at this stage (*Figure 5B*; see also *Figure 2*). We extracted RNA from four additional, independent transgenic embryos to measure changes in gene expression. Consistent with Gata6 functional binding at *Myocd* and *Jag1* enhancers, we observed a significant increase in *Myocd* and *Jag1* levels in the BA2 of all transgenic embryos overexpressing *Gata6* (*Figure 5G*). Concomitantly, we detected a marked upregulation of SMC-specific markers in the BA2 of all *a2::Gata6* embryos, which displayed a visible ectopic artery in the BA2 (3 out of 4; *Figure 5G*). These results show that forced *Gata6* expression in the NC is sufficient to initiate SMC differentiation and causes the persistence of AAs that regress under normal conditions. The presence of SMCs in all the transgenic embryos with persistent AA2 suggests that the primary effect of GATA6 is to generate SMCs and that the presence of these cells can, in most cases, stabilize the AA2 and promote its preservation.

## Discussion

The mature arterial tree emerges from the selective regression, survival and reorganization of pre-existing embryonic vessels. Migration of the cardiac NC is a vital requirement for this process. However, how the cardiac NC instructs remodeling and specifies particular AAs for survival has been unclear. Here we identify the TF Gata6 as a crucial regulator of NC differentiation to vascular SMC fates. We find that, in normal embryos, the distribution of Gata6 in the branchial area mirrors colonization by cardiac NC, as it is highly abundant in BAs3-6 and essentially absent from the BAs1-2. Additionally, we show that Gata6 is sufficient both to initiate SMC development and to support the survival of those AAs that are normally not populated by cardiac NC, and regress at early stages of

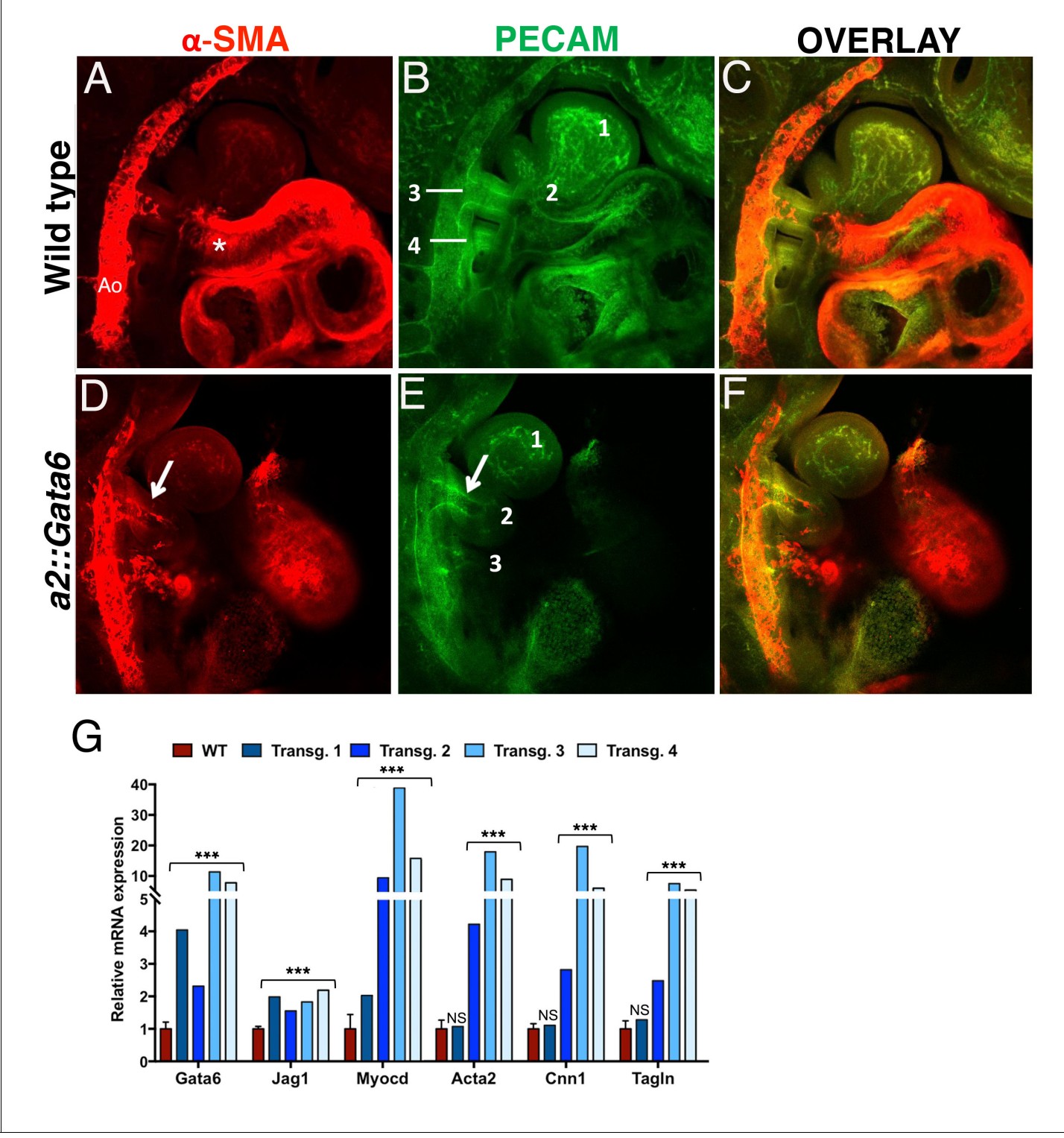

**Figure 5.** Gata6 is sufficient to induce SMCs in BA2 and to preserve AA2. Confocal analysis of whole mount wild type (**A–C**) and *a2::Gata6* transgenic mouse embryos (**D–F**) at E10.5 (s ≥ 32) to visualize vascular SMCs (a-SMA, red) (**A, D**), the endothelial network (PECAM, green) (**B, E**) and SMA/PECAM overlay (**C, F**); see also *Figure 5—videos 1–4*. (**A-C**) In wild-type embryos, SMA-positive cells are visible in the AA3, the dorsal aorta and the heart, but are not detected in BA2, where AA2 has regressed to a capillary bed (see also *Figure 2*). D-F, In *a2::Gata6* embryos, vascular SMCs are detected in the BA2 (arrows in D). In the majority of transgenic embryos analyzed, the BA2 hosts a persistent artery connected to the heart (arrow in E). For best visualization of the ectopic AA2, an individual z-stack is shown in (**D–F**); the complete z-stack series is shown in *Figure 5—videos 3–4*. (**G**) Quantitative

*Figure 5 continued on next page*

*Figure 5 continued*

RT-PCR analysis of changes in gene expression in *a::Gata6* embryos (s ≥ 37) and littermate controls. The wild-type values are presented as a mean ± SEM of two technical replicates of four independent embryos. For each transgenic embryo, the average of two technical replicates ± SEM is shown. Transgenic embryos 2–4 showed evidence of a functional AA2. Conversely, transgenic embryo one did not show significant upregulation of SMC-markers relative to wild-type; this embryo had no evidence of ectopic AA2. Asterisks (***) correspond to pvalues < 0.005; NS = not significant. All the results of transgenic embryos analyses are summarized in *Figure 5—source data 1*. Ao, dorsal aorta; numbers indicate BAs and corresponding AAs. Asterisks label the OFT.

DOI: https://doi.org/10.7554/eLife.31362.010

The following video, source data, and figure supplement are available for figure 5:

**Source data 1.** Summary of transgenic embryos results.
DOI: https://doi.org/10.7554/eLife.31362.012
**Source data 2.** Data file for *Figure 5G*.
DOI: https://doi.org/10.7554/eLife.31362.013
**Figure supplement 1.** Gata6 ectopic expression in *a2::Gata6* embryos.
DOI: https://doi.org/10.7554/eLife.31362.011
**Figure 5—video 1.** Reconstruction of wild-type embryo, stained by PECAM.
DOI: https://doi.org/10.7554/eLife.31362.014
**Figure 5—video 2.** Reconstruction of wild-type embryo, stained by a-SMA.
DOI: https://doi.org/10.7554/eLife.31362.015
**Figure 5—video 3.** Reconstruction of a transgenic *a2::Gata6* embryo, stained by PECAM to reveal the persistent AA2.
DOI: https://doi.org/10.7554/eLife.31362.016
**Figure 5—video 4.** Reconstruction of a transgenic *a2::Gata6* embryo, stained by a-SMA to reveal the ectopic SMCs in BA2.
DOI: https://doi.org/10.7554/eLife.31362.017

AA reorganization. We propose that Gata6 instructs cardiac NC to generate SMCs, and that the subsequent association of those SMCs with the caudal AAs selects these vessels for reorganization into the mature circulation (*Figure 6*).

## GATA TFs and control of vascular SMCs development

A likely possibility, based on our results, is that the distribution of SMCs within the BA area is a direct effect of Gata6 dichotomic expression, and its restriction to the cardiac NC (migrating into BAs 3–6). Positive auto-regulatory loops are a common feature of gene regulatory network. Gata6 occupancy at *Gata6* and *Gata4* loci, together with the high levels of *Gata4, Gata5* and *Gata6* transcripts in the PBA/OFT, suggest that GATA TFs regulate their own genes. However the mechanism that initiates and restricts expression of GATA TFs to the cardiac NC is not clear. The BA2 appears to be a non-permissive environment for cardiac fate (*Shenje et al., 2014*). Secondary heart field (*Kelly et al., 2001*) progenitor cells only differentiate into cardiac cells when they migrate out of the BA2, or when epithelial-free BA2 cells are grown in vitro, suggesting that signals within the BA2 support the proliferation of cardiac precursor and prevent their differentiation. An intriguing possibility is that the same signals inhibit expression of GATA TFs, which are essential for cardiac development (*Pikkarainen et al., 2004*), in the NC cells that populate the anterior BA2, and block SMC fate.

Sequence analysis predicted GATA TFs binding at regions with high Meis binding signal in the PBA/OFT. Enrichment of GATA signature motifs at these regions was accompanied by a significantly higher expression of *Gata4, Gata5* and *Gata6* in PBA/OFT relative to BA2. Computing binding levels of general regulators, combined with sequence analysis, is a novel approach which successfully identified tissue-specific instances of combinatorial binding. TFs collaborate in competing with nucleosome to gain access to underlying DNA (*Spitz and Furlong, 2012*). Increased Meis binding at regions enriched in different binding motifs across the BAs is the highly likely effect of changes in abundance of TFs, resulting in tissue-specific combination of TFs cooperating (directly or indirectly) with Meis for chromatin access. Through annotation of active enhancers and analysis of Gata6 binding in the BAs, we confirm that GATA are critical TFs driving PBA-specific transcription. Active enhancers bound by Gata6 cluster around genes with critical roles in the remodeling of the AAs and the differentiation of SMC, including *Edn1, Tgfb2, Notch1, Jag1, Myocd* (*High et al., 2007*; *Kim et al., 2013b*; *Kurihara et al., 1995*; *Manderfield et al., 2012*; *Molin et al., 2002*; *Wang et al., 2006*). Genomic regions bound by Gata6 at the *Myocd* and *Jag1* loci display active,

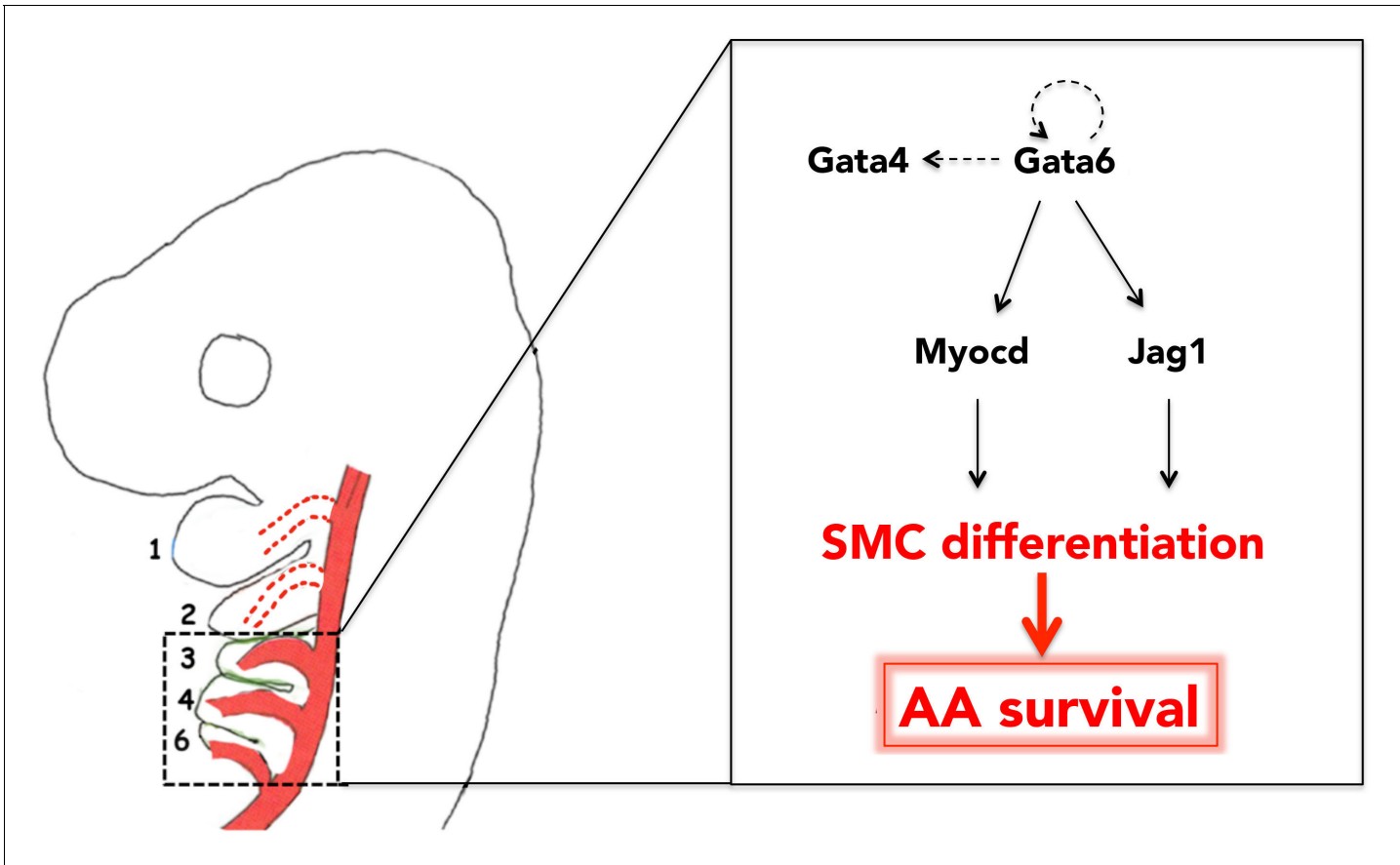

**Figure 6.** Gata6-driven SMC differentiation promotes survival of the posterior AAs. Gata6 activates *Myocd* and *Jag1* to initiate SMC differentiation in the cardiac NC, migrating to BA3-6. Generation of SMCs in BA3-6, and their recruitment by AA3-6, results in the stabilization of these vessels and their subsequent incorporation into the mature aortic tree. Conversely, because Gata6 is not expressed in BA1-2, AA1-2 do not 'see' any SMC and eventually regress. Auto-regulatory loops involving GATA TFs are likely features of Gata6 network. Arrows and double arrow indicate regulation and interaction, respectively; broken arrows indicate links that are not experimentally validated.

DOI: https://doi.org/10.7554/eLife.31362.018

tissue-restricted activity in vivo, which requires intact GATA motifs. In addition, Gata6 overexpression in BA2 leads to increased *Myocd* and *Jag1* levels. These results indicate that Gata6 acts (at least in part) via direct activation of *Myocd* and *Jag1*, to promote differentiation of SMCs.

GATA TFs control the development and homeostasis of diverse systems including blood and cardiac. We observed high enrichment of the TEAD TFs recognition motif in Gata6 peaks. The same motif also co-occurred with GATA recognition sequences at promoters of Gata6-bound enhancers. TEAD TFs control *Myocd* expression (*Creemers et al., 2006*) and can cooperate with GATA-like TFs to control gene expression (*Peng et al., 2004*), however GATA/TEAD cooperation has not been linked to vascular or SMC development before. Establishing the GATA/TEAD module as a novel regulatory feature of GATA transcriptional networks will require mapping TEAD TFs occupancy in the PBA/OFT.

*Gata6* is highly expressed in vascular SMC in mouse embryonic and postnatal development (*Morrisey et al., 1996*). However, due to conflicting data, the physiological role of Gata6 in vascular SMC is still unclear. While Gata6 has been suggested to induce and maintain the contractile SMC phenotype (*Chang et al., 2003*; *Xie et al., 2015*), additional studies have questioned a direct role of Gata6 role in the differentiation of vascular SMCs (*Lepore et al., 2005*). Most importantly, targeted inactivation of *Gata6* in SMCs and in the cranial NC leads to AAs and OFT defects without obvious abnormalities in vascular SMC development (*Lepore et al., 2006*). Here we show that ectopic *Gata6* expression in the cranial NC is sufficient to direct differentiation to SMC fates. Our observation that

Gata6 functions, at least in part, via activation of *Myocd* and Notch signaling, places Gata6 high upstream in the development of SMCs. Such an early role, possibly at the level of lineage decision, may be difficult to detect in cells that are already differentiated. Compensatory mechanisms can also explain the conflicting results observed. In our transcriptomic analyses enrichment of *Gata6* in PBA/OFT was accompanied by an equally high enrichment of transcripts encoding for other GATA family members, *Gata4 and Gata5.* In support of functional redundancy within the GATA family, Gata4 and Gata6 act cooperatively to regulate heart development and SMCs differentiation in vivo (*Xin et al., 2006*).

Mutations in GATA6 cause PTA (*Kodo et al., 2009*) and haploinsufficiency of *JAG1* (a Gata6 target detected in this study) causes Alagille syndrome, a congenital disorder associated with OFT and great vessels defects (*Li et al., 1997*; *Oda et al., 1997*). This set of newly identified Gata6 enhancers represent an important resource to map functional non-coding regions of the genome, whose genetic variants may lead to congenital heart disorders.

## NC differentiation to SMCs and the remodeling of the great vessels

We have identified Gata6 as a key factor promoting initial survival of AAs during the early stages of reorganization of the embryonic aortic arch tree. How does Gata6 support survival of these vessels? A large body of evidence indicates a striking correlation between the presence of SMCs and the fate of the AAs. During normal embryonic development vessels that will be retained, and later incorporated into the mature circulation (AA3-6), become associated with SMCs, while vessels that regress (AA1-2) do not. This correlation extends to a number of experimental conditions affecting normal remodeling of the AAs. For instance, in *Edn1* and *Ednra* mutant embryos, AA2s form correctly, but fail to regress and are wrapped by SMCs (*Kim et al., 2013b*). Similarly, we have shown here that forced Gata6 expression in BA2 NC cells promote AA2 survival and that this active vessel is also associated with SMCs. Conversely, in *Hoxa3* null mutant embryos AA3 fail to recruit SMCs and exhibit bilateral degeneration by E11.5 (*Kameda, 2009*). Likewise, in NC- ablated chick embryos AA3 formed normally, but became progressively more misshapen (*Waldo et al., 1996*). In embryos with right-sided aortic arch (in which right, rather than left, AA4-6 are selected for reorganization into the arch of the aorta and the ductus arteriosus), the distribution of SMCs changed to follow the mature fate of each artery (*Jiang et al., 2000*). Thus, SMCs specifically associate with vessels selected for preservation and subsequent incorporation into the mature heart circulation. In this study, we show that preservation of AA2 is associated with ectopic SMCs in BA2. We cannot entirely exclude that development of SMCs is secondary to the stabilization of AA2 induced by Gata6. However, we observed ectopic SMCs in all embryos expressing Gata6, including the ones where the AA2 regressed as in wild-type embryos, suggesting that the primary effect of Gata6 ectopic expression consists in SMCs development, and that SMCs subsequently promote maturation and stabilization of the vessels. Recruitment of mural cells is an essential step in the maturation of blood vessels. Mural cells form a muscular wall, which protect endothelium-lined vessels against rupture or regression. They also stabilize nascent vessels by suppressing endothelial proliferation and migration and stimulating the production of extracellular matrix (*Jain, 2003*). Indeed, blocking the interaction of mural cells with endothelial cells results in loss of vessels integrity and excessive regression of the retina vasculature (*Benjamin et al., 1998*). Together, these observations support a model in which association of vascular SMCs with the wall of the AAs is a major determinant of the physiological preservation of the posterior AAs (and its absence of the regression of anterior AAs). SMCs form the tunica media, which acts to support vessel integrity under hemodynamic load and is especially critical for the major arteries (the vessels that derive from the AAs). Consistently, the availability of SMCs and their association with the AAs could act as a major determinant in selecting those vessels that will be reorganized into the mature aortic tree.

Changes in blood flow lead to asymmetric remodelling of the embryonic circulation, and regression of right AA6 (*Yashiro et al., 2007*). We cannot exclude a contribution of flow changes (possibly caused by the rapid growth of the head region) to the regression of the anterior AAs. Nevertheless, our results support a model in which activation of a regionally restricted transcriptional program, controlled by Gata6, determines the survival of posterior AAs and their incorporation into the mature circulation. By extension, our findings demonstrate that transcriptional control is a major determinant in selecting which embryonic vessels form the mature circulation, independently from blood flow (*Yashiro et al., 2007*).

# Materials and methods

## Key resources table

| Reagent type (species) or resource | Designation | Source or reference | Identifiers | Additional information |
|---|---|---|---|---|
| strain, strain background (*Mus musculus*) | mouse CD1 | Other | Other | University of Manchester Mouse Facility |
| strain, strain background (*Danio rerio*) | zebrafish | Other | Other | University of Massachusetts Medical Center zebrafish Facility |
| antibody | anti-PECAM-1 (rat monoclonal) | BD Pharmigen | BD Pharmigen: 550274 | (1:100) |
| antibody | anti-actin, α-SMA, Cy3 conjugated (mouse monoclonal) | Sigma-Aldrich | Sigma-Aldrich: C6198 | (1:400) |
| antibody | anti-rat IgG Alexa 488-conjugated (goat polyclonal) | Molecular probes | Molecular probes: A-11006 | (1:100) |
| antibody | anti-Gata6 (rabbit monoclonal) | Cell signalling | Cel signalling: 5851 | IF: (1:1000); ChIP: (5 µg) |
| antibody | anti-Meis1/2 (goat polyclonal) | Santa Cruz Biotechnology | Santa Cruz: sc-10599X | ChIP: (5 µg) |
| antibody | anti-H3K27ac (rabbit polyclonal) | Abcam | Abcam: ab4729 | ChIP: (5 µg) |
| recombinant DNA reagent | Minitol2-GwB-zgata2-GFP-48 | | | Provided by JL Skarmeta |
| recombinant DNA reagent | Myocd-CRE2 | This paper | Other | |
| recombinant DNA reagent | Myocd-CRE1 | This paper | Other | |
| recombinant DNA reagent | Jag1-CRE1 | This paper | Other | |
| recombinant DNA reagent | Myocd-CRE2mut | This paper | Other | |
| recombinant DNA reagent | Myocd-CRE1mut | This paper | Other | |
| recombinant DNA reagent | Jag1-CRE1mut | This paper | Other | |
| recombinant DNA reagent | Gata6 cDNA | Origene | MC219384 | |
| recombinant DNA reagent | a2::Gata6 | This paper | Other | |
| recombinant DNA reagent | Meis ISH probe | Other | Other | Provided by D. Schulte |
| sequence-based reagent | Actb | Primerbank | 6671509a1 | |
| sequence-based reagent | Gata6 | Primerbank | 33859556a1 | |
| sequence-based reagent | Acta2 | Primerbank | 6671507a1 | |
| sequence-based reagent | Tagln | Primerbank | 6755714a1 | |
| sequence-based reagent | Cnn1 | Primerbank | 1069993a1 | |
| sequence-based reagent | Jag1 | Primerbank | 7305197a1 | |
| sequence-based reagent | Myocd | Primerbank | 21553083a1 | |
| sequence-based reagent | Jag1-CRE1 forward | This paper | Other | AAATCACTGTCATAATTGTCCCAAA |
| sequence-based reagent | Jag1-CRE1 reverse | This paper | Other | TCAGGGCTTCCCACTGCTA |
| sequence-based reagent | Myocd-CRE1 forward | This paper | Other | TGCATGCTGCCCCCATCAAT |
| sequence-based reagent | Myocd-CRE1 reverse | This paper | Other | GAGGCGCAACCCAATGT GC |
| sequence-based reagent | Myocd-CRE2 forward | This paper | Other | TCTGGACAGCTGACACCCTTG |
| sequence-based reagent | Myocd-CRE2 reverse | This paper | Other | TGAGCAATAAGGGACAGGAGC |
| commercial assay or kit | Gateway BP Clonase II Enzyme mix | Thermo Fisher Scientific | 11791020 | |

| commercial assay or kit | pCR8/GW/TOPO TA Cloning Kit with One Shot TOP10 E. coli | Thermo Fisher Scientific | 450642 | |
|---|---|---|---|---|
| commercial assay or kit | TruSeq ChIP Sample Preparation kit | Illumina | | |
| commercial assay or kit | TruSeq Stranded mRNA Sample Preparation Kits | Illumina | | |
| commercial assay or kit | QIAquick Gel Extraction Kit | Qiagen | 28704 | |
| commercial assay or kit | RNeasy Plus Micro Kit | Qiagen | 74034 | |
| chemical compound, drug | Methyl Salicylate | Sigma-Aldrich | M2047 | |
| chemical compound, drug | Trizol | Thermo Fisher Scientific | 15596–018 | |
| software, algorithm | Amira software | FEI | Version 6.4. | |
| software, algorithm | MACS | *Zhang et al., 2008* | MACS2.0.10. 20131216 | https://pypi.python.org/pypi/MACS2/2.0.10.20131216 |
| software, algorithm | Trimmomatic | *Bolger et al., 2014* | v0.32 | http://www.usadellab.org/cms/?page=trimmomatic |
| software, algorithm | Bowtie | *Langmead and Salzberg, 2012* | Bowtie(v1.1.1)/ Bowtie(v2.2.3) | https://sourceforge.net/projects/bowtie-bio/files/bowtie2 |
| software, algorithm | GREAT | *McLean et al., 2010* | Other | http://bejerano.stanford.edu/great/public/html/ |
| software, algorithm | TopHat | *Kim et al., 2013a* | TopHat (v2.1.0) | https://github.com/infphilo/tophat |
| software, algorithm | Cufflinks | *Trapnell et al., 2010* | Cufflinks(v2.2.2) | https://github.com/cole-trapnell-lab/cufflinks |
| software, algorithm | edgeR | *Robinson et al., 2010* | edgeR (v3.12.1) | https://bioconductor.org/packages/release/bioc/html/edgeR.html |
| software, algorithm | Homer | *Heinz et al., 2010* | Other | http://homer.salk.edu/homer/ |
| software, algorithm | DiffReps | *Shen et al., 2013* | Other | https://code.google.com/p/diffreps/under |

## Animals

Wild-type (CD1) mice were time-mated to obtain embryos for microdissections. For experiments, embryos were not selected for gender and were used between E10.5 and E11.5.

Wild-type zebrafish were raised in the Animal Facility at the University of Manchester and in the University of Massachusetts Medical Center Zebrafish Facility.

Experiments on animals followed the local (ASPA 1986, UK; Portaria 1005/92 and Directive 2010/63/EU, P) legislations concerning housing, husbandry, and welfare.

## Generation of mouse transgenic embryos

The *a2::Gata6* transgenic construct contained the BA2 enhancer of the *Hoxa2* gene (*Nonchev et al., 1996*) linked to a minimal promoter (*Kanzler et al., 1998*), cloned upstream of the Gata6 cDNA (Origene) and a SV40 polyadenylation signal. The construct was gel purified using the QIAquick gel extraction kit (Qiagen, UK) and used to produce transgenic embryos by pronuclear injection (*Hogan et al., 1994*). Embryos were genotyped by PCR on yolk sac genomic DNA, using oligonucleotides designed on the *Hoxa2* promoter and *Gata6* cDNA. We obtained a total of 11 transgenic embryos, seven were processed for confocal imaging and four for RNA extraction. Embryos were recovered at E10.5, fixed in 4% paraformaldehyde overnight at 4°C and taken to methanol through a graded methanol PBT (PBS containing 0.1% Tween 20) series and stored at −20°C until further processed for imaging. For gene expression analysis, embryos were isolated at E10.5 and BA2 were dissected from individual embryos and snap frozen in dry ice. RNA was extracted from individual BA2 pairs using RNeasy Plus Micro Kit (Qiagen), transcribed into cDNA and analysed using StepOne-Plus Real_Time PCR Systems (Life Technologies) and PrimerBank primers (*Spandidos et al., 2010*). To compute the statistical significance of the qPCR data we used Zscore transformation. Zscores

were calculated by subtracting the average of $2^{-\Delta\Delta Ct}$ values of each gene in the WT from the $2^{-\Delta\Delta Ct}$ value of the corresponding gene in the transgenic, and then dividing the result by the standard deviation of all $2^{-\Delta\Delta Ct}$ values for the gene in WT. The Zscores thus obtained were converted into pvalues using the Rpackage *pnorm*.

## Generation of zebrafish transgenic embryos

Embryos and adults zebrafish were maintained under standard laboratory conditions. Enhancers were amplified from mouse genomic DNA using the primers (listed in Key resources table), cloned into pCR8/GW/TOPO vector (Life Technologies) and recombined using the Gateway system (Life Technologies) to an enhancer test vector that includes a strong midbrain enhancer (Minitol2-GwB-zgata2-GFP-48, a kind gift from JL Skarmeta) as an internal control. The mutant enhancers were generated by Genscript in a pUC57 vector, PCR amplified and cloned into Minitol2-GwB-zgata2-GFP-48 as described above. Zebrafish embryos were collected from natural spawning. The plasmid DNA was injected into the cytoplasm of embryos at the one-cell stage. Injected embryos were visualized at 48 hr post fertilization using a Leica fluorescent stereomicroscope. Fisher Exact test was performed to test the changes in activity of the wildtype and mutant enhancers.

## ChIP-seq and downstream analyses

CD1 mice were time-mated to obtain branchial arches and OFT from E115 embryos. ChIP-seq has been described in detail (*Donaldson et al., 2012*). The antibodies used were: Gata6 (Cell Signaling, RRID:AB_10705521), Meis (Santa Cruz Biotechnology, RRID:AB_2143020) and H3K27ac (Abcam, RRID:AB_2143020). Following Chromatin-immunoprecipitation (ChIP), DNA libraries were constructed using the TruSeq ChIP Sample Preparation Kit (Illumina, Inc.). The final purified product was sequenced on an Illumina HiSeq2500 instrument. For histone ChIP-seq, single-end reads were trimmed to 50 base pairs (bp) and mapped to mouse reference sequence (mm9/NCBI37) using Bowtie v1.0.0 (RRID:SCR_005476). Peak calling was done using MACS2.0 (*Zhang et al., 2008*) (RRID:SCR_013291). For Gata6 and Meis ChIP-seq, read pairs (R1 and R2) were filtered using Trimmomatic v0.32 (*Bolger et al., 2014*) (RRID:SCR_011848) using paired-end mode, to remove adapters, and truncated reads (3') with a base sequence quality of <Q20. Filtered reads <50 bp were removed. Filtered paired reads were mapped to mouse reference sequence (mm9/NCBI37) using Bowtie2 v2.2.3 with default parameters. Mapped paired-reads were filtered with 'samtools' v0.1.19, to remove reads with mapping quality <Q30 and discordant pairs (i.e. incorrect orientation and/or > 500 bp apart). Peak calling was done using MACS 2 v2.1.0. Binding regions were reported with a minimum q-value of 0.05. For all ChIP-seq only paired reads belonging to chromosomes 1–19, X and Y were used in downstream analyses. Narrow peaks (TF ChIP-seq) were filtered using FE ≥ 10. Each experiment was performed in duplicate; for H3K27Ac and Meis ChIP-seq, two replicates were intersected and peaks present in both replicates were used in downstream analyses. For Gata6 replicate 1 was used for downstream analysis; replicate 2 resulted in fewer binding regions with FE ≥ 10 (=693), which were almost entirely contained in replicate 1 (97%). Details of all ChIP-seq experiments (e.g. number of reads, etc) are provided in *Supplementary file 4*. ChIP-seq peaks were associated to genes using GREAT (version 2.0.2; RRID:SCR_005807) (*McLean et al., 2010*) (http://great.stanford.edu/) and the 'basal plus extension' association rule. The comparison of genome coordinates used GALAXY (*Goecks et al., 2010*). De novo motif discovery was done using the '*findMotifGenome*' module of the HOMER package (*Heinz et al., 2010*) (RRID:SCR_010881) on Gata6 peaks with FE ≥ 10, extended 100 bp from the summit position in each direction. DiffReps (*Shen et al., 2013*) (RRID:SCR_010873) was used to detect differential binding of Meis across PBA/OFT and BA2. Meis ChIP-seq experiments in PBA/OFT and BA2 (binding locations) were compared using a 200 nt sliding window. Regions with logFC ≥3 binding in PBA versus BA2 were selected and intersected with Meis ChIP-seq 200nt summits in PBA, FE ≥ 10, for de novo motif discovery using HOMER. The workflow is summarized in *Supplementary file 5*.

Heatmaps for Gata6 and H3K27ac peaks in PBA/OFT and BA2 were generated using the histogram matrix produced by '*annotatePeaks*' module of HOMER package and an R script. For each dataset, peaks were extended to 5 kb from the summit in each direction. The analysis of gene annotation enrichment was performed using GREAT and the 'basal plus extension' association rules. The ratio of the H3K27ac signal for up and down regulated gene in PBA/OFT and BA2 was computed

and plotted against the ratio of their corresponding gene expression values (FPKM) using the RNA-seq data. H3K27ac signals for all DE genes were represented by the average RPM (reads per million mapped reads) values of all peaks annotated to the respective gene. For all upregulated (or downregulated) genes, average RPM of their associated regions in PBA (or BA2) were divided by the average RPM values of the corresponding regions in BA2 (or PBA).

## RNA-seq and downstream analyses

Branchial arches and OFT were dissected and snap frozen in dry ice. RNA was extracted using Trizol (Life Technologies), pooling 3–5 pairs of BAs for each sample. Embryos with 38–39 (TS17) and 48–50 somites (TS19) were used for E10.5 and E11.5 time points, respectively. RNA-seq was performed in duplicate and the quality of duplicates assessed using PCA. Library preparation of samples was performed using the Illumina TruSeq Stranded mRNA Sample Preparation Kit (Illumina). Fastq files for each tissue were mapped against the mouse genome (mm9) using the spliced aligner Tophat (v2.1.0, RRID:SCR_013035) (*Kim et al., 2013a*) with default parameter settings. rRNA transcripts were removed from the mapped bam files. Details of all RNA-seq experiments (e.g. number of reads, etc) are provided in *Supplementary file 4*. Expression levels for each tissue were quantified using Cuffdiff program in the Cufflinks package (v2.2.2; RRID:SCR_014597; RRID:SCR_001647) (*Trapnell et al., 2013*), which estimates the raw counts and FPKM values for each gene. FPKM values were used for the correlation plot to assess the global expression profile. The raw counts from the cuffdiff were subsequently used to quantify the differential expression levels for genes using the package edgeR (v3.12.1; RRID:SCR_012802) (*Robinson et al., 2010*). Multiple testing correction was done using the R package qvalue (*Storey et al., 2015*). LogFC of the genes was plotted against their $\log_{10}$ (qvalues) as a volcano plot to give an overview of significant DE genes. Hierarchical clustering of different tissues at the two time points was performed using $\log_2$ transformed CPM values and heatmaps plotted using the R package hclust and heatmap2. Clustering was performed employing Euclidean distance as the similarity metric and average linkage as the clustering method.

## Immunofluorescence and in situ hybridization

Immunofluorescence on whole mount embryos was adapted from previously described protocols (*Foo et al., 2006*). Embryos were incubated overnight at 4°C with the primary antibody, rat anti-mouse PECAM-1 (BD Pharmingen, RRID:AB_393571, 1:100) or Gata6, washed and incubated overnight at 4°C with α-SMA, Cy3-conjugated (Sigma-Aldrich, RRID:AB_476856) and goat anti-rat IgG Alexa 488-conjugated (Invitrogen, RRID:AB_141373, 1:100). After washing, embryos were dehydrated and mounted in Methyl Salicylate (Sigma-Aldrich, M2047) for clearing. Images were collected on a Leica TCS SP5 AOBS inverted confocal microscope and processed using ImageJ (v1.48). Amira software (Version 6.4; FEI, RRID:SCR_014305) was used to create the reconstructions of complete z-stacks of wild-type and transgenic embryos. Sagittal sections of paraffin embedded E10.5 embryos were labeled using the antibodies described above. In situ hybridization was carried out as described previously (*Kanzler et al., 1998*), using *Meis1* probe (a gift from Dorothea Schulte).

## Data availability

The ChIP-seq and RNA-seq experiments have been submitted to ArrayExpress. (accession numbers: E-MTAB-5407, E-MTAB-5536 and E-MTAB5394).

# Acknowledgements

We thank Stuart Marshall for experimental support, Shane Herbert for critical reading of the manuscript, Ian Donaldson and the members of the Genomic Technologies and Bioinformatics, Imaging, Histology and Biological Services Core Facilities at the University of Manchester, Jose Luis Gomez Skarmeta for the Gateway-Z48-GFP vector. This work was supported by MRC grant MR/L009986/1 to NB and NAH, BBSRC grant BB/N00907X/1 to NB and a BBSRC studentship to ML. FL and CS were supported by NIH grant NS038183 to CS.

## Additional information

### Funding

| Funder | Grant reference number | Author |
| --- | --- | --- |
| Medical Research Council | MR/L009986/1 | Nicoletta Bobola |
| Biotechnology and Biological Sciences Research Council | BB/N00907X/1 | Nicoletta Bobola |
| National Institute of Neurological Disorders and Stroke | NS038183 | Charles Sagerström |

The funders had no role in study design, data collection and interpretation, or the decision to submit the work for publication.

### Author contributions

Marta Losa, Conceptualization, Investigation, Writing—original draft, Writing—review and editing; Victor Latorre, Franck Ladam, Ana Novoa, Laure Bridoux, Investigation, Writing—original draft, Writing—review and editing; Munazah Andrabi, Peyman Zarrineh, Data curation, Investigation, Writing—original draft, Writing—review and editing; Charles Sagerström, Supervision, Funding acquisition, Investigation, Writing—original draft, Writing—review and editing; Neil A Hanley, Conceptualization, Funding acquisition, Writing—original draft, Writing—review and editing; Moises Mallo, Conceptualization, Supervision, Investigation, Writing—original draft, Writing—review and editing; Nicoletta Bobola, Conceptualization, Supervision, Funding acquisition, Investigation, Writing—original draft, Writing—review and editing

### Author ORCIDs

Charles Sagerström, http://orcid.org/0000-0002-1509-5810
Neil A Hanley, http://orcid.org/0000-0003-3234-4038
Moises Mallo, https://orcid.org/0000-0002-9744-0912
Nicoletta Bobola, http://orcid.org/0000-0002-7103-4932

### Ethics

Animal experimentation: Experiments on animals followed the local (ASPA 1986, UK; Portaria 1005/92 and Directive 2010/63/EU, P) legislations concerning housing, husbandry, and welfare.

### Decision letter and Author response

Decision letter https://doi.org/10.7554/eLife.31362.035
Author response https://doi.org/10.7554/eLife.31362.036

## Additional files

### Supplementary files

• Supplementary file 1. DE genes in PBA and BA2
DOI: https://doi.org/10.7554/eLife.31362.020

• Supplementary file 2. Meis differential binding in PBA and BA2
DOI: https://doi.org/10.7554/eLife.31362.021

• Supplementary file 3. Gata6 peaks and overlap with H3K27Ac peaks
DOI: https://doi.org/10.7554/eLife.31362.022

• Supplementary file 4. RNA-seq and ChIP-seq statistics
DOI: https://doi.org/10.7554/eLife.31362.023

• Supplementary file 5. Flowchart of the RNA-seq and ChIP-seq downstream analyses
DOI: https://doi.org/10.7554/eLife.31362.024

• Transparent reporting form
DOI: https://doi.org/10.7554/eLife.31362.025

## Major datasets

The following datasets were generated:

| Author(s) | Year | Dataset title | Dataset URL | Database, license, and accessibility information |
|---|---|---|---|---|
| Marta Losa, Nicoletta Bobola | 2017 | RNA-seq analysis of branchial arches and outflow tract of the mouse embryo at E10.5 and E11.5 | http://www.ebi.ac.uk/arrayexpress/experiments/E-MTAB-5394 | Publicly available at EBI (accession no: E-MTAB-5394) |
| Marta Losa, Laure Bridoux, Nicoletta Bobola | 2017 | ChIP-seq for Meis on mouse branchial arches at E11.5 | http://www.ebi.ac.uk/arrayexpress/experiments/E-MTAB-5536 | Publicly available at EBI (accession no: E-MTAB-5536) |
| Marta Losa, Victor Latorre, Nicoletta Bobola | 2017 | ChIP-seq for Gata6 and histone H3K27Ac on mouse branchial arches at E11.5 | http://www.ebi.ac.uk/arrayexpress/experiments/E-MTAB-5407 | Publicly available at EBI (accession no: E-MTAB-5407) |

The following previously published datasets were used:

| Author(s) | Year | Dataset title | Dataset URL | Database, license, and accessibility information |
|---|---|---|---|---|
| He A, Gu F, Ma Q, Ye L, Visel A, Pennacchio LA, Pu WT | 2014 | Reinstatement of developmental stage-specific GATA4 enhancers controls the gene expression program in heart diseaseGata4 ChIP-seq | https://www.ncbi.nlm.nih.gov/geo/query/acc.cgi?acc=GSE52123 | Publicly available at the NCBI Gene Expression Omnibus (accession no: GSE52123) |
| Minoux M, Vitobello A, Kitazawa T, Kohler H, Stadler MB, Rijli FM | 2017 | Gene bivalency at Polycomb domains regulates cranial neural crest positional identity [ATAC-seq] ATAC-seq BA2 | https://www.ncbi.nlm.nih.gov/geo/query/acc.cgi?acc=GSE89436 | Publicly available at the NCBI Gene Expression Omnibus (accession no: GSE89436) |

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
