## [Decision Letter]

[Editors’ note: a previous version of this study was rejected after peer review, but the authors submitted for reconsideration. The first decision letter after peer review is shown below.]

Thank you for submitting your work entitled "A tissue-specific, Gata6-driven transcriptional program instructs remodeling of the mature arterial tree" for consideration by *eLife*. Your article has been reviewed by three peer reviewers, and the evaluation has been overseen by a Senior Editor. The reviewers have elected to remain anonymous.

Our decision has been reached after consultation between the reviewers. Based on these discussions and the individual reviews below, we regret to inform you that your work will not be considered further for publication in *eLife*.

As you will see from the full reviews, appended below, the reviewers find your paper very interesting and potentially appropriate for *eLife*. However, they also raised many concerns that would need to be dealt with for the manuscript to be considered further. Because *eLife* has a policy that papers are only considered for revision if they can be revised within a two month time-frame. The consensus was that the required changes would take considerably longer. Therefore, we cannot accept the paper at this time but would encourage you to make the required changes and submit a new paper that addresses the reviewers' comments. We would then make every effort to return the paper to the same reviewers. I hope you find the reviews helpful in revising the manuscript.

*Reviewer #1:*

The manuscript by Losa et al. describes a transcriptional program that is driven by GATA-6 and instructs the remodeling of the arterial tree, as indicated in the title. The results show that the neural crest cells differentiate into vascular smooth muscle cells only around aortic arches that are going to survive and reorganize. This process is driven by the transcriptional program driven by GATA-6.

This is a very interesting paper that however suffers from some bias that are not necessarily introduced by the AA, but are present in most of the literature on Chipseq and similar technologies. However these bias do not affect the final conclusions of the manuscript nor the results of the reported experiments, I feel. However, I believe that they should be taken care of. The fact that several other AA draw conclusions in a certain way does not mean that their conclusion is correct.

I have the following observations.a) The authors have used Meis1 binding sites to detect tissue-specific combinatorial TF occupancy. They should more carefully explain the basis of this choice.

b) The AA use data-base consensus sequences to identify, for example, Meis1 binding sites. In particular they use a six bp consensus. This is not correct. Indeed, this consensus is used by other transcription factors of the same family and, in particular, by the Pbx1-Meis1 transcription dimer. This dimer, however, also binds to the same consensus that the AA identify as Hox-Pbx. This is not a mistake as this binding site indeed binds ALSO Hox-Pbx. However, the AA's choices may in part bias the results on the identification of other TF, or at least of their selectivity.

I don't believe that these problems affect the results presented in the paper. However, the literature, in particular the data banks, are too full of ill-conceived data, like those on some consensus binding sites. The AA do not mention wherefrom they obtain the used consensus sites. I stress the importance of the use of the data obtained in actual experiments rather than those from data banks.

*Reviewer #2:*

Losa et al. provided evidence for the identification of a GATA transcriptional program in posterior branchial arch tissues that is required for the survival and reorganization of aortic arches. Analysis of ChIP-seq data allows the characterization of GATA6 peaks at the jag1 and Myocd loci. The authors hypothesize that regulation of these genes would promote development of the aortic arches into mature bloods vessels.

In general, this study is an interesting investigation combining ChIP-seq data and in vivo validations. This is interesting because GATA6 is a critical factor for cardiovascular development and that remodeling of branchial aortic arches is still unclear. Yet, I remain unconvinced as to the role of GATA6-Jag1/Myocd in the branchial arches could regulate the survival and reorganization of these tissues. While, the regulation of Jag1 and Myocd by GATA6 is likely, the mechanistic insights from this paper remain incomplete and could benefit from a more expanded analysis.

1) Rostro-caudal progression of branchial arches depends on a developmental timing of the embryogenesis. The choice of developmental stages E10.5 and E11.5 for a comparative analysis does not allow a perfect differential study. The authors should indicate why they used these stages for their RNA-seq analysis.

2) In Figure 1, the authors comment variation of transcripts expressed in SMCs. However, several other transcripts expressed in myocardium are highly enriched, which could reflect the presence of OFT in their comparative analysis. The authors should discuss the limitation of their study.

3) Vascular persistence of the most anterior aortic arches could be associated with the formation of the head, which reduce the pressure of blood in this region. Yashiro et al. (2007) have demonstrated that blood pressure is responsible for remodeling of the great arteries. The authors should discuss how haemodynamics could influence genetic program during the formation of aortic arches.

4) The use of Meis binding to detect tissue-specific is intriguing. The author should better justify their choice, especially because their model proposed at the end does not include such 'ubiquitous' factor.

5) Motif for GATA transcription factors was among the most highly overexpressed sequences but was not alone since other Forkhead, T-box or TEAD motifs were also enriched, together with Meis motifs. The authors use a shortened to jump to GATA without clear arguments.

6) Sequence for the TEAD family of transcription factors was often associated with Gata6 peaks (Figure 4 and Figure 4—figure supplement 1). Again, the author did not explore the hypothesis where a GATA/TEAD module is important feature of the posterior aortic arch transcriptional network.

7) The authors used zebrafish to validate the Cis Regulatory Element identified in Myocd and jag1 loci. However, Data presented are not convincing, particularly in terms of images quality. Please clarify whether the CREs are expressed in zebrafish vascular system.

8) Gata6 mutant should be used to validate whether jag1 and Myocd are regulated by this factor.

9) a2:Gata6 embryos were used as a model of gain of function. The authors should use these embryos to show an up-regulation of jag1 and Myocd. Persistence of AA2 is not obvious in the embryos showing in Figure 5 and Figure 5—figure supplement 1). Quality of images should be increased.

10) "The presence of SMCs in all the transgenic embryos with persistent AA2 suggests that the primary effect of Gata6 is to generate SMCs and that the presence of these cells can, in most cases, stabilize the AA2 and promote its survival." This sentence is confusing, if the AA2 survivals, that implies that there is less apoptosis. It does not seem that this mechanism was involved in the persistence of the AA2.

*Reviewer #3:*

This is a very nicely presented paper that uses a combination of embryo imaging, transcriptome analysis, ChIP-seq analysis and in vivo mutagenesis (mouse and fish) to identify a critical role for Gata6 in controlling smooth muscle cell (SMC) differentiation from neural crest cells to support the developing posterior arch arteries. Importantly the authors demonstrate that it is the recruitment of these SMC that is crucial for the maintenance of the posterior arch arteries so that they can persist and remodel into the aortic arch arteries, whereas the anterior arches, without recruitment of SMC, are able to regress. Forced expression of Gata6 in Hox-expressing neural crest cells causes the anterior arch arteries to persist and become invested in SMC.

The sequencing and bioinformatics analysis all appears to have been conducted to a very high standard.

There are some considerations, however, with the embryo work that need to be addressed.

The development of the arch arteries (AA) in the mouse is a rapid process with the formation and regression of the first 2 AA occurring within the E9.5 stage, and the remodelling stage occurring during E11.5. Therefore it is important that the authors accurately describe the stage of embryo development, particularly within E9.5, at analysis. This is conventionally done by somite counting and should be achievable from the images already acquired. The authors should also state how well stage-matched the embryos are within the experimental groups, and this should be done for all experiments where embryos are used (immunostaining and sequencing). It is well known that embryos from the same litter can be at different stages of development as determined by somite number.

Only whole-mount images of stained embryos (immuno and in situ) are presented. Although the images are of high quality it is not always obvious which tissue type or layer is being stained. Confirmation can only be achieved through double immunostaining of sections, or sectioning of the in situ stained embryos. This would clarify the location of the cells stained and give confidence to the conclusions drawn from these images. In particular this would be useful for the Gata6/SMA stained embryos shown in Figure 3 as it is not clear in which cells Gata6 is expressed in the 4th AA. If it is not smooth muscle cells, could it be in NCC? Can this be demonstrated? Would using an alternative antibody, e.g. SM22alpha, label SMC around the 4th PAA at this stage?

[Editors’ note: what now follows is the decision letter after the authors submitted for further consideration.]

Thank you for submitting your article "A tissue-specific, Gata6-driven transcriptional program instructs remodeling of the mature arterial tree" for consideration by *eLife*. Your article has been reviewed by three peer reviewers, and the evaluation has been overseen by Marianne Bronner as the Senior Editor and Reviewing Editor. The reviewers have opted to remain anonymous.

The reviewers have discussed the reviews with one another and the Reviewing Editor has drafted this decision to help you prepare a revised submission.

Summary:

This much improved manuscript investigates the genetics underlying development of the connection of the heart with systemic circulation. Using several embryonic species, the authors conclude that preservation of some (not all) embryonic vessels results from differentiation of neural crest cells into smooth muscle cells around those vessels that are due to be preserved and reorganized. They identify Gata 6 as crucial in this process. Gata 6 is expressed in smooth muscle cells and drives their differentiation through MyoCD and Jag1. They also identify Meis1 as an important player in the process since it allows Gata 6 binding through a chromatin effect.

Essential revisions:

1) The manuscript needs some rewriting as it is difficult to read because the "real" ChIP-seq peaks cannot be derived from data with a threshold q<0.05 which is not sufficiently stringent compared to papers using comparable techniques. For example, this results in the identification, of 50,000 Meis1 peaks whereas, using a more stringent threshold one gets 5,000. In the end, the authors only use those peaks and genes which have a high fold enrichment. Are the conclusions based on "all" peaks or only on the subset with high fold enrichment? The authors should clarify this and are encouraged to reevaluate their data using higher stringency.

2) The reader needs a clearer view of the actual data. A table stating the total number of reads, number of identified peaks (or reads), statistical thresholds, should be presented.

---

## [Author Response]

[Editors’ note: the author responses to the first round of peer review follow.]

Reviewer #1:[…] This is a very interesting paper that however suffers from some bias that are not necessarily introduced by the AA, but are present in most of the literature on Chipseq and similar technologies. However these bias do not affect the final conclusions of the manuscript nor the results of the reported experiments, I feel. However, I believe that they should be taken care of. The fact that several other AA draw conclusions in a certain way does not mean that their conclusion is correct.I have the following observations.a) The authors have used Meis1 binding sites to detect tissue-specific combinatorial TF occupancy. They should more carefully explain the basis of this choice.

The same point has been raised by reviewer 2; we thank both reviewers for the opportunity to provide additional information to clarify the rationale of this approach.

The approach we have used is based on the observation that two or more TFs (binding to relatively short stretches of DNA) can compete with the repositioning of nucleosome on DNA, while TF binding in isolation are not so efficient in doing that. Based on this phenomenon, or ‘collaborative DNA binding’, we expect binding of TFs to DNA to become more efficient (i.e. higher binding levels) in the event of combinatorial binding. We would also expect combinatorial binding (and the binding motifs recognized) to reflect concentration of TFs in the nucleus, i.e. to be tissue-specific. Based on the above, changes in the binding levels of any TF across multiple tissues (provided the TF of choice is present in multiple tissues), could be used as ‘sensors’ to detect tissue-specific instances of TF combinatorial occupancy. Meis is expressed in both BA2 and PBA. Most importantly, Meis binding is very widespread (Meis TFs bind to tens of thousands of regions) and lends itself very well to the interrogation of a large portion of the genome.

We have added a few explanatory sentences to the Results section (subsection “3. Identification of the main regulators of the PBA/OFT transcriptional network”, first paragraph), to explain this approach. To support the choice of Meis as a widespread ‘sensor’ for collaborative binding, we analysed ATAC-seq data obtained from E10.5 BA2. We found an extensive overlap between Meis binding and ATAC-seq, with Meis occupying 60% of accessible chromatin regions in the BA2. This overlap is quite remarkable when considering the different experimental approaches used (ATAC-seq vs Meis ChIP-seq) and the different embryonic stage (E10.5 vs E11.5). This analysis has been added to Figure 3—figure supplement 1.

b) The AA use data-base consensus sequences to identify, for example, Meis1 binding sites. In particular they use a six bp consensus. This is not correct. Indeed, this consensus is used by other transcription factors of the same family and, in particular, by the Pbx1-Meis1 transcription dimer. This dimer, however, also binds to the same consensus that the AA identify as Hox-Pbx. This is not a mistake as this binding site indeed binds ALSO Hox-Pbx. However, the AA's choices may in part bias the results on the identification of other TF, or at least of their selectivity.

We entirely agree that it is important to use motifs derived from actual experiments. Whenever possible (in most cases), we have used motifs identified in ChIP-seq experiments (references provided in Figure 3—figure supplement 1). Specifically, the six letter Meis motif was identified by de novo motif discovery as the top enriched motifs in Meis ChIP-seq in the BA2 and in the PBAs.

I don't believe that these problems affect the results presented in the paper. However, the literature, in particular the data banks, are too full of ill-conceived data, like those on some consensus binding sites. The AA do not mention wherefrom they obtain the used consensus sites. I stress the importance of the use of the data obtained in actual experiments rather than those from data banks.Reviewer #2:[…] In general, this study is an interesting investigation combining ChIP-seq data and in vivo validations. This is interesting because GATA6 is a critical factor for cardiovascular development and that remodeling of branchial aortic arches is still unclear. Yet, I remain unconvinced as to the role of GATA6-Jag1/Myocd in the branchial arches could regulate the survival and reorganization of these tissues. While, the regulation of Jag1 and Myocd by GATA6 is likely, the mechanistic insights from this paper remain incomplete and could benefit from a more expanded analysis.1) Rostro-caudal progression of branchial arches depends on a developmental timing of the embryogenesis. The choice of developmental stages E10.5 and E11.5 for a comparative analysis does not allow a perfect differential study. The authors should indicate why they used these stages for their RNA-seq analysis.

We carefully considered including earlier time points over a developmental time course to allow comparing earlier BA2 stages versus later PBAs stages. As part of the rostro-caudal progression mentioned by the reviewer, new AAs are being formed posteriorly. The area corresponding to the PBAs changes over time, with the addition of new arteries from E9.5 until E10.5. Specifically, AA4 and AA6 appear at E10.0 and E10.5 respectively. These changes in the PBA tissues are not ideal for the BA2/PBA comparison (whose main aim is to identify transcriptional changes between NC-colonized areas where vessels regress to NC-colonized areas where vessels persist). Taking this into account, we decided to use E10.5 as the earliest time stage, because the PBA is ‘complete’, and E11.5 as the latest stage, because shortly after (E12.0) the aortic arches lose their left-right symmetrical organization. To take into account rostro-caudal progression we compared late PBA (E11.5) to early BA2 (E10.5), but this did not change the results, presumably because differences are already established at the earliest time point (Figure 1).

2) In Figure 1, the authors comment variation of transcripts expressed in SMCs. However, several other transcripts expressed in myocardium are highly enriched, which could reflect the presence of OFT in their comparative analysis. The authors should discuss the limitation of their study.

The enrichment of myocardium transcripts is indeed another feature of the PBA/OFT tissue. It is most likely caused by the presence of the OFT, included to preserve the physical integrity of the AA3-6 and also as a landmark to allow dissection of comparable material across different embryos. While this feature does not prevent the identification of differences between anterior and posterior aortic arches, we agree with the reviewer that is important to clarify this point.

The following sentence has been added to the Results section:

“PBA/OFT-enriched genes also included transcripts highly expressed in the myocardium, owing to the OFT component (Figure 1—figure supplement 1).”

3) Vascular persistence of the most anterior aortic arches could be associated with the formation of the head, which reduce the pressure of blood in this region. Yashiro et al. (2007) have demonstrated that blood pressure is responsible for remodeling of the great arteries. The authors should discuss how haemodynamics could influence genetic program during the formation of aortic arches.

This is indeed a valid suggestion; we have discussed a potential role for this mechanism at the end of the Discussion.

4) The use of Meis binding to detect tissue-specific is intriguing. The author should better justify their choice, especially because their model proposed at the end does not include such 'ubiquitous' factor.

The approach we have used is based on the observation that two or more TFs (binding to relatively short stretches of DNA) can compete with the repositioning of nucleosome on DNA, while TF binding in isolation are not so efficient in doing that. Based on this phenomenon, or ‘collaborative DNA binding’, we expect binding of TFs to DNA to become more efficient (i.e. higher binding levels) in the event of combinatorial binding. We would also expect combinatorial binding (and the binding motifs recognized) to reflect concentration of TFs in the nucleus, i.e. to be tissue-specific. Based on the above, changes in the binding levels of any TF across multiple tissues (provided the TF of choice is present in multiple tissues), could be used as ‘sensors’ to detect tissue-specific instances of TF combinatorial occupancy. Meis is expressed in both BA2 and PBA. Most importantly, Meis binding is very widespread (Meis TFs bind to tens of thousands of regions) and lends itself very well to the interrogation of a large portion of the genome.

We have added a few explanatory sentences to the Results section (subsection “3. Identification of the main regulators of the PBA/OFT transcriptional network”, first paragraph), to explain this approach. To support the choice of Meis as a widespread ‘sensor’ for collaborative binding, we analysed ATAC-seq data obtained from E10.5 BA2. We found an extensive overlap between Meis binding and ATAC-seq, with Meis occupying 60% of accessible chromatin regions in the BA2. This overlap is quite remarkable when considering the different experimental approaches used (ATAC-seq vs. Meis ChIP-seq) and the different embryonic stage (E10.5 vs E11.5). This analysis has been added to Figure 3—figure supplement 1.

5) Motif for GATA transcription factors was among the most highly overexpressed sequences but was not alone since other Forkhead, T-box or TEAD motifs were also enriched, together with Meis motifs. The authors use a shortened to jump to GATA without clear arguments.

We have rephrased the text to explain the focus on GATA TFs:

“The most highly overrepresented motifs identified by de novo motifs analysis, matched recognition sequences for GATA, Forkhead, HAND and TEAD, together with motifs recognized by Meis and Meis partners Hox/Pbx (Figure 3). […] Consistent with changes in combinatorial occupancy reflecting changes in protein concentration, *Gata6,* together with *Gata4* and *Gata5,* wasamong the most significantly enriched transcripts in PBA/OFT (Figure 1—figure supplement 1).”

6) Sequence for the TEAD family of transcription factors was often associated with Gata6 peaks (Figure 4 and Figure 4—figure supplement 1). Again, the author did not explore the hypothesis where a GATA/TEAD module is important feature of the posterior aortic arch transcriptional network.

We have explored the potential role of the GATA/TEAD module in the posterior aortic arch transcriptional network. First, we calculated the co-occurrence of TEAD and GATA motifs in Gata6 peaks. We found that the occurrence of TEAD and GATA (most Gata peaks contain a GATA motif) is much more frequent than expected by chance (pval<e-39, see table below) and suggests that TEAD and GATA may indeed bind together in this embryonic area. Within Gata peaks, the average distance of the GATA-TEAD motifs is about 40nt (Author response image 1). Gata6 peaks, containing GATA and TEAD motifs, cluster around genes involved in heart valve development and import of nuclear proteins, most notably SMAD (Author response image 2). We also found that several genes essential for aortic arch development (e.g. Bmps, SMAD, Jag1) are associated to Gata6 peaks containing GATA-TEAD motifs. While these observations are interesting, and suggestive of a functional binding interaction, their significance in the context of aortic arch development remains unclear. A rigorous characterization of the GATA/TEAD network would require in vivo binding of TEAD in the same (or a closely related) embryonic tissue as GATA. More specifically, because Gata6 peaks containing GATA-TEAD motifs do not map to the enhancers analyzed in Figure 4, we feel that, within the current Results section, this additional analysis may seem a bit incongruous. For the reasons above, we would prefer not to include this analysis in the new version of the manuscript. We have slightly modified the TEAD-related section in the Discussion (subsection “GATA TFs and control of vascular SMCs development”) to stress the importance of analyzing in vivo TEAD binding to reach clear conclusions. For the same reasons, we have removed TEAD from the model in Figure 6.

**Author response image 2. respfig2:** 

Frequencies of co-occurring Gata:TEAD motifs in Gata6 peaks

7) The authors used zebrafish to validate the Cis Regulatory Element identified in Myocd and jag1 loci. However, Data presented are not convincing, particularly in terms of images quality. Please clarify whether the CREs are expressed in zebrafish vascular system.

We used transgenesis in zebrafish to establish regulation of Jag1 and Myocd enhancers by GATA TFs in vivo. We measured transgenes’ activities in hundreds of embryos to generate robust, statistically significant results (Figure 4—figure supplement 2). In order to do this, we generated transient transgenic embryos and scored the same primarily injected animals for GFP activity (driven by the wild-type and mutant Jag1 and Myocd enhancers) in the heart/OFT. The heart/OFT is partly related to the PBA/OFT and, when analyzing many embryos, GFP signal is easier and more consistent to score as positive in the heart/OFT compared to vessels.

Because transient transgenesis is characterized by mosaic expression of the transgene, a careful investigation of the transgene expression pattern would require establishing stable lines. We initially established lines for Myocd CRE1, whose activity is consistent with the developing heart and vasculature. A picture of GFP reporter activity driven by Myocd CRE1 has been added to Figure 4—figure supplement 1, showing that Myocd CRE1 drives GFP in the heart and in vessels-like structures. However, as this approach is time-consuming and highly limits the numbers of embryos to be analyzed, we only generated stable lines for this construct. Subsequently, we moved to transient transgenesis to perform a more quantitative analysis of wild-type and mutant enhancer activity.

8) Gata6 mutant should be used to validate whether jag1 and Myocd are regulated by this factor.9) a2:Gata6 embryos were used as a model of gain of function. The authors should use these embryos to show an up-regulation of jag1 and Myocd. Persistence of AA2 is not obvious in the embryos showing in Figure 5 and Figure 5—figure supplement 1). Quality of images should be increased.

We thank the reviewer for this excellent suggestion, which has significantly improved the manuscript. We have chosen to use a gain of function system to address this request and quantified the levels of Jag1 and Myocd transcripts in transgenic embryos overexpressing Gata6 in the BA2. We generated new a2-Gata6 transgenic embryos, isolated BA2 from individual embryos, extracted RNA and quantified transcripts using qPCR. We found a significant increase in Myocd and Jag1 transcripts in the BA2 of all embryos overexpressing Gata6 compared to wild-type. The increase in Jag1 and Myocd levels, following Gata6 overexpression in BA2, reflect their differential expression in PBA versus BA2. Specifically, Jag1, whose levels are around 3 times higher in PBA vs BA2, is upregulated around 2 fold in transgenic embryos. Myocd transcript levels, which are over 50 fold higher in PBA versus BA2, are generally increased to much higher levels in transgenic embryos. The expression variability across transgenic embryos (relative to wild-type embryos) is consistent with each embryo having an independent transgene integration site. These data have been added to Figure 5.

We believe that upregulation of Jag1 and Myocd transcripts in response to Gata6 overexpression, together with Gata6 binding at functional enhancers within Myocd and Jag1 loci and the requirement of intact GATA motifs for enhancers’ activity, conclusively establishes Jag1 and Myocd as Gata6 direct targets.

We also used the same experiment to quantify the expression of vascular SMC-specific markers in the BA2 of transgenic embryos. Notably, the 3 transgenic embryos (2-3-4), which displayed blood in the BA2, a sign of a patent artery, had significantly higher levels of Acta2, Cnn1 and Tagln1 in the BA2 relative to wild-type, while embryo 1 (which did not show blood in the BA2 and presumably had no ectopic artery) showed similar levels of these transcripts to wild-type embryos. These data provide additional support to the ectopic differentiation of SMC in the BA2 of a2-Gata6 embryos and complete the analysis presented in Figure 5.

Regarding point 8, we have serious reservations that Gata6 mutants would not conclusively address the functional link between Gata6 and Myocd and Jag1 genes. Targeted inactivation of *Gata6* in SMCs and in the cranial NC leads to AAs and OFT defects without obvious abnormalities in vascular SMC development (Lepore et al., 2006). The relatively ‘normal’ process of vascular SMC differentiation in these mutants is likely to reflect relatively healthy levels of its upstream regulators, i.e. Jag1 and Myocd. We have discussed possible redundancy with Gata4 and /or Gata5, which are also highly expressed in the PBAs similar to Gata6, as a possible explanation.

To best visualize the persistent AA2, we have modified Figure 5 and added Videos 1-4. To improve the quality of the images, we modified Figure 5 by replacing the entire zstack with a smaller subset of z-stack (n=15), which shows the ectopic AA2 more clearly; the entire zstack is shown in Videos 3-4, along with Video 1-2 (wild-type).

10) "The presence of SMCs in all the transgenic embryos with persistent AA2 suggests that the primary effect of Gata6 is to generate SMCs and that the presence of these cells can, in most cases, stabilize the AA2 and promote its survival." This sentence is confusing, if the AA2 survivals, that implies that there is less apoptosis. It does not seem that this mechanism was involved in the persistence of the AA2.

We agree with the reviewer; survival has been replaced with preservation in the last sentence of the subsection “5. Forced expression of GATA6 in the neural crest generates SMCs”.

Reviewer #3:[…] There are some considerations, however, with the embryo work that need to be addressed.The development of the arch arteries (AA) in the mouse is a rapid process with the formation and regression of the first 2 AA occurring within the E9.5 stage, and the remodelling stage occurring during E11.5. Therefore it is important that the authors accurately describe the stage of embryo development, particularly within E9.5, at analysis. This is conventionally done by somite counting and should be achievable from the images already acquired. The authors should also state how well stage-matched the embryos are within the experimental groups, and this should be done for all experiments where embryos are used (immunostaining and sequencing). It is well known that embryos from the same litter can be at different stages of development as determined by somite number.

This is an important point. We have added the number of somites as follows:

RNA-seq (Figure 1): We pooled embryos with similar somites numbers for RNA-seq. Theiler stages and somite numbers added to Materials and methods section (subsection “RNA-seq and downstream analyses”).

Confocal analysis of AAs development (Figure 2): somite numbers added to figure legend.

Gata6 and Gata6/SMA Immunofluorescence (Figure 3): somite numbers added to figure legend.

Confocal and expression analysis of transgenic embryos (Figure 5): Embryos overexpressing Gata6 were obtained from two separate injections, the first resulting in embryos with > 32 somites, the second resulting in older embryos (>38 somites). In all cases embryos were too advanced to display a wild-type AA2 artery. This has been added to the figure legend.

The only experiments where somites numbers are not available are ChIP-seq. Careful selection or recording of developmental stages is impractical for these experiments, which require a high number of embryos.

Only whole-mount images of stained embryos (immuno and in situ) are presented. Although the images are of high quality it is not always obvious which tissue type or layer is being stained. Confirmation can only be achieved through double immunostaining of sections, or sectioning of the in situ stained embryos. This would clarify the location of the cells stained and give confidence to the conclusions drawn from these images. In particular this would be useful for the Gata6/SMA stained embryos shown in Figure 3 as it is not clear in which cells Gata6 is expressed in the 4th AA. If it is not smooth muscle cells, could it be in NCC? Can this be demonstrated? Would using an alternative antibody, e.g. SM22alpha, label SMC around the 4th PAA at this stage?

To clarify the location of the cells stained, we have added double immunostaining on section, using Gata6 and α-SMA antibodies (3I). The picture shows SMA-positive cells surrounding AA3, and localization of SMA to Gata6-expressing cells.

In addition we have added an individual zstack from confocal analysis (3H), which shows the presence of differentiating SMC (SMA-positive) in the proximal region of BA4, in Gata6-positive areas.

[Editors' note: the author responses to the re-review follow.]

Essential revisions:1) The manuscript needs some rewriting as it is difficult to read because the "real" ChIP-seq peaks cannot be derived from data with a threshold q<0.05 which is not sufficiently stringent compared to papers using comparable techniques. For example, this results in the identification, of 50,000 Meis1 peaks whereas, using a more stringent threshold one gets 5,000. In the end, the authors only use those peaks and genes which have a high fold enrichment. Are the conclusions based on "all" peaks or only on the subset with high fold enrichment? The authors should clarify this and are encouraged to reevaluate their data using higher stringency.

We thank the reviewers for the opportunity to clarify these issues.

As correctly mentioned by the reviewers, we have not used all peaks for downstream analyses, but only high confidence peaks with fold enrichment (FE) values higher than 10. This resulted in the elimination of a large fraction of peaks detected by MACS2. More specifically, in addition to q-value < 0.05, which is MACS2 default cutoff to call significant regions, we have applied an additional stringent cutoff of FE ≥10 to all transcription factors ChIP-seq, i.e. to ‘narrow’ peaks. Additionally, for Meis ChIP-seq downstream analysis, we have used peaks FE≥10, which were present in both replicates.

As a FE≥10 cutoff is too stringent for ‘broad’ peaks, for histone ChIP-seq (whose peaks are wider and flatter because of nucleosome distribution on chromatin), we selected peaks present in both replicates for downstream analysis.

One of the standards prior to peak calling is to ensure that the IP experiment contains more than 10 million aligned short reads, and the newly added table (see point below) shows that all ChIP experiments fulfil this pre-requirement. The other standard measurement of the quality of peaks detected in transcription factors ChIP-seq experiments is motif search. For both Meis and GATA ChIP-seq, Homer motif search highlighted a main motif with high *p*-value, and detected the main motif in most of the binding regions.

To clarify how peaks were selected for downstream analysis, we have enclosed a diagrammatic figure, which highlights the selection of FE≥10 as a step of the workflow (Supplementary file 5 in the submission). In addition, we have made the following changes to the manuscript (highlighted in the manuscript’s text):

1) Materials and methods: Narrow peaks (TF ChIP-seq) were filtered using FE≥10. Each experiment was performed in duplicate; for H3K27Ac and Meis ChIP-seq, two replicates were intersected and peaks present in both replicates were used in downstream analyses.

2) Materials and methods: Details of all ChIP-seq experiments (e.g. number of reads, etc.) are provided in Supplementary file 4.

3) Materials and methods: Regions with logFC≥ 3 binding in PBA versus BA2 were selected and intersected with Meis ChIP-seq 200nt summits in PBA, FE≥10, for de novo motif discovery using HOMER.

4) Materials and methods: The workflow is summarized in Supplementary file 5.

5) Materials and methods: Details of all RNA-seq experiments (e.g. number of reads, etc.) are provided in Supplementary file 4.

6) Figure 3, legend: Diffrep analysis of Meis binding in PBA and BA2. Venn diagram shows ‘shared’ Meis binding sites (FE≥10) with logFC<3 signal.

7) Figure 3, legend Top over-represented biological processes associated to Meis peaks (FE≥10).

8) Figure 3, legend: de novo motif discovery on Meis peaks (FE≥10) with higher signal in PBA.

9) In the previous version, the analysis presented in Figure 3—figure supplement 1 (overlap between E10.5 ATAC-seq in BA2 and Meis binding in BA2), was performed using Meis peaks FE≥5. To maintain consistency across the manuscript, we have performed a new analysis with Meis peaks FE≥10. Now all transcription factor ChIP-seq downstream analyses use FE≥10 peaks.

10) Figure 3—figure supplement 1, has been changed accordingly: Comparison of Meis ChIP-seq (FE≥10 peaks) in E11.5 BA2 and ATAC-seq in E10.5 BA2. Half of Meis peaks overlap with accessible chromatin in the BA2.

11) Figure 3—figure supplement 1, Experimentally identified motifs (known motifs), which correspond to Homer identified motifs, were used to count the instances of highly enriched (logFC<3) Meis binding (200nt summit regions, FE≥10)

2) The reader needs a clearer view of the actual data. A table stating the total number of reads, number of identified peaks (or reads), statistical thresholds, should be presented.

We thank the reviewers for the valuable suggestion. We have added a table, which contains the total number of reads, number of peaks and other relevant values for the various steps of the workflow, for all the ChIP-seq and RNA-seq experiments. This is named ‘Supplementary file 4’ in the submission.